# Newcastle disease virus promotes spreading infection through vimentin-dependent tight junction injury mediated by MLC/p-MLC activation

Xiaolong Lu[1,2,3], Qiwen Zhou[1], Mingzhu Wang[1], Meiqi Li[1], Wenhao Yang[1,2,3], Kaituo Liu[2,3,4], Ruyi Gao[1,2,3], Tianxing Liao[1,2,3], Yu Chen[1,2,3], Jiao Hu[1,2,3], Min Gu[1,2,3], Shunlin Hu[1,2,3], Xinan Jiao[1,2,3,4], Xiaoquan Wang[1,2,3], Xiufan Liu [1,2,3]*, Chan Ding[2,5]*, Xiaowen Liu[1,2,3]*

**1** Key Laboratory of Avian Bioproducts Development, Ministry of Agriculture and Rural Affairs, Yangzhou University, Yangzhou, China, **2** Jiangsu Co-Innovation Center for Prevention and Control of Important Animal Infectious Diseases and Zoonosis, Yangzhou University, Yangzhou, China, **3** Jiangsu Key Laboratory of Zoonosis, Yangzhou University, Yangzhou, China, **4** Joint International Research Laboratory of Agriculture and Agri-Product Safety of Ministry of Education of China, Yangzhou University, Yangzhou, China, **5** Shanghai Jiao Tong University School of Agriculture and Biology, Shanghai Jiao Tong University, Shanghai, China

* xfliu@yzu.edu.cn (XFL); shoveldeen@shvri.ac.cn (CD); xwliu@yzu.edu.cn (XWL)

## Abstract

Newcastle disease virus (NDV), a widespread poultry pathogen, spreads efficiently via the respiratory tract. However, the precise mechanism governing its spreading infection remains unclear. This study reveals that NDV-induced tight junction (TJ) injury is crucial for viral replication and spread. NDV infection significantly reduced TJ proteins OCLN and ZO-1 through multiple degradation pathways involving viral proteins, disrupting TJ integrity and promoting cell migration. Knockdown of OCLN and ZO-1 further enhanced viral replication and spread, underscoring their importance. Concurrently, NDV altered the distribution of OCLN and ZO-1, accompanied by cytoskeletal rearrangements of vimentin and F-actin. Notably, NDV triggered vimentin and F-actin rearrangement to form cage-like structures, benefiting TJ injury and viral replication. Critically, vimentin rearrangement was essential for the redistribution of OCLN, ZO-1, and F-actin, facilitating viral replication, spread, and inflammation. MLC/p-MLC activation was required for vimentin-mediated TJ injury, thereby promoting NDV replication and spread. Unlike avirulent strains, the virulent NDV promoted replication and spread through vimentin-mediated TJ injury, subsequently worsening lung damage in chickens. These findings elucidate how NDV rapidly disseminates and worsens lung damage, providing insights relevant to the pathogenesis and treatment of viral pneumonias, including those caused by coronaviruses and influenza viruses.

**Data availability statement:** All relevant data are within the manuscript and its Supporting information files. The raw data of this study are openly available in Figshare at https://doi.org/10.6084/m9.figshare.29527019.v2.

**Funding:** This work is supported by the National Natural Science Foundation of China (32302840 to X.L), the National Key Research and Development Project of China (2023YFD1800601 to X.L; 2023YFD1800605 to K.L), the Open Project Program of Jiangsu Key Laboratory of Zoonosis (R1808 to X.L), the Priority Academic Program Development of Jiangsu Higher Education Institutions (PAPD to X.L), the Earmarked Fund For China Agriculture Research System (CARS-40 to S.H), and the 111 Project (D18007 to X.L). The funders played no role in study design, data collection, analysis and interpretation of data, or the writing of this manuscript.

**Competing interests:** The authors have declared that no competing interests exist.

## Author summary

Newcastle disease virus (NDV) causes high poultry mortality and significant economic losses globally. Its spreading infection poses a major health risk to birds. Our study reveals a key mechanism behind NDV's rapid spread. NDV targets tight junction (TJ) proteins OCLN and ZO-1, degrading them and disrupting the junctions. Normally, F-actin binds intracellular ZO-1, which links to OCLN at the membrane, forming a strong TJ barrier. We found that vimentin rearrangement is crucial for redistributing OCLN, ZO-1, and F-actin, aiding viral replication, spread, and inflammation. Critically, NDV induces vimentin and F-actin to form cage-like structures, worsening TJ injury and promoting viral replication. Unlike the avirulent NDV, the virulent NDV exploits this mechanism to enhance viral spread and inflammation. Activation of the MLC/p-MLC pathway is necessary for vimentin-induced TJ injury, thereby boosting NDV replication and spread. Thus, we demonstrate how NDV exploits vimentin-mediated TJ disruption, via the MLC/p-MLC pathway, to rapidly replicate and spread.

## Introduction

Cells typically connect via specialized junctions, among which tight junctions (TJs) are crucial adhesion complexes that maintain the integrity of adjacent epithelial cells [1]. As the primary stabilizers of the epithelial barrier, TJs function as the front-line physical defense against pathogenic microorganisms [2]. TJ proteins form the structural framework of TJs. Key components include the transmembrane proteins occludin (OCLN) and claudin-1 (CLDN1), and the scaffolding protein zonula occludens-1 (ZO-1) [3,4]. Transmembrane proteins are arrayed in linear formations, creating "welding lines" that encircle cells, securely fusing adjacent cell membranes and effectively sealing intercellular spaces. Concurrently, cytoplasmic scaffolding proteins serve to anchor the "welding lines" to the cytoskeleton, thereby reinforcing the stability of TJs [5,6]. Vitally, cytoskeleton proteins can form TJ complexes with TJ proteins, which plays an important role in maintaining the intercellular TJ homeostasis [7]. Although non-epithelial cells are not preferred models for studying TJs, certain TJ proteins may indeed be highly expressed on the surface of these cells, which can then lead to the formation of TJ-like structures [8,9].

   Recently, a large number of research have elucidated that viruses are capable of hijacking the cellular TJs to facilitate their infection processes [10–12]. To enhance its spread, the virus disrupts TJs by downregulating TJ proteins, thus preventing their retention by opening the paracellular route [13]. Such disruption of TJs can expedite the relaxation or complete dissociation of intercellular connections. Certain viruses have adapted by targeting the integrity of intercellular tight junctions, triggering damage that enhances the shedding and migration of infected cells, thereby facilitating the propagation of viral infection [11,14]. Occasionally, the process of cell-free transmission is triggered by the viral egress from the host cell. Following this event,

the virus accesses the extracellular environment [15]. Several viruses can facilitate their spread by compromising TJs within the host, culminating in severe systemic infections [16,17]. Previous studies have identified a subset of TJ proteins involved in viral cell–cell spread, yet other molecules are equally vital for spreading infection. Recently, the host intermediate filaments protein vimentin has emerged as a key player in viral spread. It's critical for HCV infection *in vivo*, and lacking vimentin significantly hinders HCV's ability to spread between cells [18].

Newcastle disease virus (NDV) is responsible for a high incidence and mortality rate among poultry, resulting in substantial economic losses to the global poultry industry [19]. NDV encodes six structural proteins (NP, P, M, F, HN, L) and two non-structural proteins (W, V) via six open reading frames [20,21]. These viral proteins pose significant effects on multiple stages of the NDV infection cycle. Notably, the F and HN proteins are membrane glycoproteins critical for infection and pathogenicity, and the cleavage sites of NDV F protein are crucial for viral virulence [22]. NDV primarily infects hosts through the respiratory tract, with the TJ injury in respiratory epithelial cells being crucial for viral infection [23,24]. In particular, highly virulent NDV strains can rapidly disseminate to multiple tissues, culminating in systemic infection and severe viremia [25]. Our lab previously established an *in vivo* infection model for NDV, and found that the highly virulent NDV strain can rapidly induce lung injury and proliferate in the bloodstream of chickens. It then spreads throughout the body, ultimately leading to high pathogenicity and mortality in the infected chickens [26]. Hence, the effective dissemination of the virus constitutes a critical pathway through which NDV exerts its pathogenic potential. Furthermore, our previous study observed that the virulent NDV significantly induces the rearrangement of vimentin fibers, and its infectivity is closely associated with the rearranged vimentin fibers [27]. Cytoskeletal vimentin sustains cellular structural homeostasis; It collaborates with microtubules and microfilaments to establish a comprehensive cellular lattice system, and is involved in modulating the infection processes of diverse viruses [28,29]. Impairment of cellular barriers associated with TJs is conducive to viral infection [30,31]. Particularly, vimentin has been demonstrated to mediate epithelial barrier opening by regulating TJs, thereby influencing infection of pathogens [32]. Nevertheless, the precise relationship of the interplay between vimentin and TJs remains to be further elucidated. Therefore, this study aims to investigate whether NDV promotes its spread by modulating TJs and to explore the potential interaction mechanisms between NDV and TJs. This study is the first to show that NDV infection downregulates TJ protein expression and disrupts the TJ structure *in vitro* and *vivo*. We found that this TJ disruption facilitates NDV replication and spread. Furthermore, the cytoskeletal protein vimentin is involved in this TJ damage, with accompanying rearrangement of both vimentin and F-actin. This effect occurs in both mammalian and avian cells, but only after infection with the virulent NDV strain, not the avirulent one. Importantly, activation of MLC/p-MLC is required for NDV-induced TJ injury, thereby promoting viral replication and spread. Taken together, this study deciphered the potential mechanism behind viral dissemination and acute lung injury by NDV and offers a fresh perspective on the pathogenesis and intervention strategies for viral pneumonias, such as those caused by the novel coronavirus and influenza virus.

## Materials and methods

### Ethics statement

All animal experiments were conducted in strict accordance with the guidelines outlined in the Regulations on the Administration of Laboratory Animals by the State Scientific and Technological Commission of the People's Republic of China. The Yangzhou University Experimental Animal Ethics Committee (Permission number: 202302083) approved all of the animal studies according to the Administrative Measures of Jiangsu Province on laboratory Animals of Jiangsu Administrative Committee of Laboratory Animals.

### Viruses, animals, cells, and antibodies

NDV strains including genotype II (La Sota), IV (Herts/33), and IX (F48E8) were provided by our laboratory. Specific pathogen-free (SPF) chicken embryos and chickens were purchased from Beijing Merial Company (Beijing, China) and

Jinan Speifului Poultry Technology Co., Ltd (Shandong, China), respectively. The biological characteristics of NDV strains were shown in S1 Table. The viruses were propagated in the allantoic cavity of 10-day-old SPF chicken embryos. Following a 48-hour (h) incubation period, the allantoic fluid was collected and preserved at −70°C for subsequent analysis and characterization. Human lung epithelial (A549) cells, chicken macrophage (HD11) cells, human cervical cancer epithelial (HeLa) cells, chicken tracheal epithelial (CTE) primary cells, and chicken embryo fibroblasts (CEFs) were preserved and cultured in our laboratory. The half-maximal tissue culture infective dose ($TCID_{50}$) of NDV was subsequently determined. To achieve inefficient viral replication, viral suspensions were irradiated with ultraviolet (UV) light as described previously. Viral inactivation was then confirmed by validating the lack of replication in 10-day-old chicken embryonated eggs and in CEF monolayer cultures. The antibodies used in this study were as follows: mouse anti-NDV NP (provided by our laboratory), rabbit anti-OCLN, rabbit anti-ZO-1, rabbit anti-CLDN1 (CST, USA), rabbit anti-MLC, rabbit anti-p-MLC (Affinity, China), mouse anti-vimentin (Proteintech, China), Actin-Tracker Red-594 (Beyotime, China), anti-mouse or anti-rabbit HRP IgG (CWBIO, China), anti-mouse or anti-rabbit FITC (TransGen, China) or AF594 IgG (Bioss, China).

## Cytopathic observation and cell viability

Cytopathic effects were observed utilizing a cellular morphology assay under light microscopy. Briefly, cells seeded in 6-well plates were either mock-infected or inoculated with NDVs at a multiplicity of infection (MOI) of 1. The morphological changes in each treatment group were monitored at specified time points using a microscope at 100 × magnification. Cell viability was assessed using the Cell Counting Kit-8 (CCK-8, Beyotime Biotech, China) according to the manufacturer's instructions. For this purpose, cells were cultured in 96-well plates and infected with NDVs at an MOI of 1 for defined durations. Post-infection, the cells were maintained at 37°C for 1h after the addition of 10 μL of CCK-8 reagent. Subsequently, the absorbance at 450 nm for each well was quantified using an enzyme-linked immunosorbent assay (ELISA) reader.

## Transwell assay

Cell migration was assessed using a 24-well Transwell assay (Corning Incorporation, Corning, NY, USA). Cells seeded in 6-well plates were infected with NDV at an MOI of 0.1 for either 6 or 24h. Subsequently, the infected cells were detached using trypsin. A suspension of quantitatively prepared cells (detailed preparation method described in the respective figure legend) in 200 μL of serum-free medium was then added to the upper chamber of the Transwell insert (pore size, 8 μm). The lower chamber was filled with 500 μL of complete culture medium supplemented with 20% fetal bovine serum (FBS) to create a chemoattractant gradient. After a 36-hour incubation at 37°C, the membrane was washed with PBS to remove non-migratory cells remaining on the upper surface. Migratory cells that had transmigrated to the underside of the membrane were fixed with 4% paraformaldehyde. Following fixation, these migrated cells were stained with crystal violet. For quantification, five randomly selected fields per membrane were imaged under an inverted light microscope, and the number of migratory cells in each field was counted [33].

## Transepithelial electrical resistance (TEER) assay

A549 cells were cultured to 90% confluence and dissociated into a single-cell suspension using trypsin. The suspension was plated at $2 \times 10^5$ cells/500 μL in the upper compartment of a transwell insert, with 1.5 mL of fresh medium in the lower compartment. Cells were allowed to adhere and form a monolayer barrier. NDV was then added at 0.1 MOI, with a control group untreated. The TEER measurement (Millicell, Germany) was taken at specific time points post-infection to assess barrier function. Resistance per unit area was calculated as "Resistance per unit area = Resistance (Ω) × Effective membrane area ($cm^2$)".

## Western blot assay

Cells were lysed in RIPA buffer (Beyotime Biotech, China) supplemented with the protease inhibitor PMSF (Beyotime Biotech, China) at the indicated time. The protein content of the lysates was quantified utilizing the BCA Protein Assay Kit (Beyotime

Biotech, China). Subsequently, the proteins were denatured and fractionated via 10% SDS-PAGE, followed by transferring onto polyvinylidene difluoride (PVDF) membranes. After blocking, the membranes were subjected to incubation with appropriately diluted primary and secondary antibodies. Protein detection was achieved by treating the membranes with a chemiluminescent substrate and imaging using a chemiluminescence imager (Tanon, China) in a darkroom environment. Lastly, the optical density of the protein bands was quantified using ImageJ version 1.48v software (National Institutes of Health, USA).

## Plasmid construction and transfection

To generate plasmids expressing His-tagged viral NP, P, V, W, M, F, and HN proteins, as well as a Flag-tagged L protein, the coding sequences for each gene were amplified using gene-specific primer pairs and cDNA derived from the F48E8 strain (GenBank Accession Number: FJ436302) as the template, with sequences provided in S2 Table. These amplified fragments were subsequently cloned into the XbaI/BamHI or EcoRI/XbaI restriction sites of the pcDNA3.1-His/Flag vector using the Basic Seamless Cloning and Assembly Kit (TransGen, China), following the manufacturer's protocol. Successful plasmid construction was confirmed via DNA sequencing. All plasmids were named pcDNA3.1-gene name-tag, including pcDNA3.1-NP-His, pcDNA3.1-P-His, pcDNA3.1-M-His, pcDNA3.1-F-His, pcDNA3.1-HN-His, pcDNA3.1-V-His, pcDNA3.1-W-His, and pcDNA3.1-L-Flag.

For transfection experiments, cells were seeded to achieve 70% confluency prior to transfection with the respective plasmids, employing the TransIntro EL Transfection Reagent (TransGen, China) according to the manufacturer's instructions.

## RNA interference and pharmacological treatment

Small interfering RNAs (siRNAs) targeting TJ genes was synthesized by GenePharma (China), with sequences provided in S3 Table. A non-targeting siRNA (siRNA-NC) was employed as a negative control. Cells cultured in 24-well plates were transfected with 60 pmol of siRNA using 2 µL of EL Transfection Reagent (TransGen Biotech, China), following the transfection protocol provided by the manufacturer. The efficiency of siRNA transfection was confirmed by western blot analysis at 24 hours post-transfection (hpt). Thereafter, cells exhibiting successful gene knockdown were selected for subsequent experimental investigations.

Cells were infected with NDV at the indicated dose at 37°C. Following a 1-h adsorption period, the cells were rinsed with PBS and subsequently cultured in DMEM supplemented with 1.5% 3,3'-iminodipropionitrile (IDPN) (Sigma-Aldrich, USA), 20 µM Z-VAD-FMK, 20 µM MG132 (Beyotime, China), and 1 µM Bafilomycin A1 (TOPSCIENCE, China) for the duration indicated. Furthermore, cells were treated with 10 µM ajoene (GLPBIO, USA) for 12 h prior to infection with NDV at the indicated dose for the specified duration. Subsequent to the incubation period, the cells were prepared for further analysis.

## Confocal microscopy assay

Cells grown on 14-mm coverslips were treated with siRNAs or pharmacological agents prior to infection with NDV at the indicated dose for the specified durations. Subsequent to the infection period, the cells were washed with PBS, fixed with 4% paraformaldehyde, permeabilized using Triton X-100 (Beyotime Biotech, China), and blocked with 5% bovine serum albumin (BSA). The primary antibody was then added and incubated overnight at 4°C, followed by a 1-h incubation at 37°C with either FITC-conjugated or AF594-conjugated secondary antibodies. Afterward, nuclear staining was performed with DAPI (Beyotime Biotech, China), and the coverslips were mounted using PBS supplemented with 50% glycerol. The prepared samples were finally visualized using a confocal microscope (Leica, Germany).

## Transmission electron microscopy (TEM) assay

Cells were infected with NDV at an MOI of 0.1 for 6 h. Following infection, the cells were removed from the culture medium, subjected to a PBS wash, and then incubated with electron microscopy fixative at room temperature in the dark for a duration

of 5 min. The cells were gently detached in a unidirectional manner using a cell scraper and aspirated into a centrifuge tube. Following centrifugation at 2000 rpm for 5 min, the cell sediment was resuspended in fresh electron microscopy fixative. An additional 30-min fixation was conducted at room temperature in the dark, after which the sample was transferred to 4°C for preservation. The prepared cells were subsequently sent to Servicebio (Wuhan, China) for TEM examination and imaging.

**Animal experiments**

NDVs were inoculated into six 4-week-old SPF chickens at a dose of $10^5$ 50% egg infectious dose (EID$_{50}$)/0.1mL per chicken via the intranasal and intraocular routes, respectively. Control chickens received an equivalent volume of PBS. At 4 days post-inoculation (dpi), chickens from each group were humanely euthanized, and lung tissue was harvested to assess histopathological alterations. The lung samples were fixed with neutral formalin solution, processed through a series of dehydration and paraffin embedding steps, sectioned, baked, dewaxed, and stained with hematoxylin and eosin. Following the preparation of histological sections, the histopathological modifications were examined and documented using a light microscope. Furthermore, 0.3 g of lung tissue was homogenized in RIPA, followed by centrifugation. The supernatant was then subjected to determine the protein expression of OCLN and ZO-1. Additionally, oropharyngeal and cloacal swabs were gathered daily from the infected chickens to evaluate virus shedding. The swab samples were meticulously scrubbed and subjected to centrifugal force. Subsequently, they were used to inoculate 9-day-old SPF chicken embryos. At 96 hpi, allantoic fluid was harvested and evaluated using the hemagglutination (HA) assay for analysis.

**RNA extraction and real-time quantitative PCR (RT-qPCR)**

For RNA extraction and subsequent cDNA synthesis, TransZol Up Reagent (TransGen Biotech, China) and PrimeScript RT Reagent (Takara, China) were employed, respectively. The mRNA expression was quantified using reverse transcription quantitative polymerase chain reaction (RT-qPCR). To assess the amplification of target genes, 2 μL of cDNA from each sample was amplified using the SYBR Green dye-based method (Vazyme, China), adhering to the manufacturer's instructions. Relative gene expression was determined by the $2^{-\Delta\Delta CT}$ method. The results are presented as the fold change ($2^{-\Delta\Delta CT}$) in log10 relative to the control group.

To determine the expression of TJ genes, cells in 12-well plates were subjected to mock infection or inoculated with 1 MOI of NDV for 24 h. To measure the levels of inflammatory cytokines, cells treated with IDPN or ajoene were subjected to NDV infection for 24 h, and 0.3g of above-mentioned lung tissue was harvested at 4 dpi. Subsequently, RNA was extracted from the treated cells and lung tissue, followed by reverse transcription to cDNA. The mRNA expression levels were then assessed using qPCR. The TJ and inflammatory genes encompassed OCLN, ZO-1, IL-1β, IL-6, IL-18, and TNF-α, with the primer sequences detailed in S4 Table.

**Immunofluorescence assay (IFA)**

Cells treated with specific siRNAs or pharmacological agents were exposed to NDVs at the indicated dose for the specified duration. Post-infection, the cells underwent a series of processing steps including PBS washing, fixation with 4% paraformaldehyde, permeabilization, and blocking with 5% BSA. Subsequently, the cells were subjected to an overnight incubation at 4°C with an anti-NP primary antibody, followed by a 1-hour incubation at 37°C with a secondary antibody. Fluorescence microscopy was employed to visualize and photograph the cells, with nuclear staining performed using DAPI. For quantitative analysis, fifteen random lesions per treatment group were chosen, and the extent of viral spread was determined by counting the number of cells within each lesion.

**Plaque assay**

Cells were infected with NDV at a low dose (0.00001 MOI) at 37°C for 1 h. Subsequently, the cells were rinsed with PBS to eliminate any unattached viruses and overlaid with a 3% low-melt agarose gel. Once the overlay agar had solidified, the

PLOS Pathogens

plate was inverted and maintained at 37°C for incubation. Following 60 h of incubation, the cells were subjected to staining with crystal violet (Beyotime, China). The relative plaque size was quantified using ImageJ software.

## Statistical analysis

The data are presented as the means ± standard deviations of three independent replicates derived from a representative experiment. Either one-way or two-way analysis of variance (ANOVA) was utilized to evaluate the statistically significant differences, with a significance threshold set at $p < 0.05$. The statistical analyses were conducted with GraphPad Prism 7.00 (San Diego, USA).

## Results

### NDV disrupts the TJ homeostasis with the reduced expression of TJ-associated proteins

We first established an NDV infection model in the A549 cell line and monitored cellular morphology at various time intervals post-infection. The control and avirulent NDV LaSota groups maintained characteristic cell morphology throughout the observation period. In contrast, infection with the virulent NDV strains Herts/33 and F48E8 resulted in pronounced cellular damage, characterized by enhanced intercellular spaces, vacuolation, cell rounding, aggregation, and even disintegration. Notably, the F48E8 strain elicited the most pronounced degree of cellular damage across all infected groups, with the significant cellular lesion observed at 12 hpi (Fig 1A). Tight junction is the main mode of connection between epithelial cells, and the findings of cellular damage imply that NDV infection may affect intercellular tight junction integrity. In general, impairment of tight junction integrity can enhance cellular migration [11]. Here, we employed a transwell assay to determine the cell migration ability induced by NDV (Fig 1B). The results showed that both the virulent NDV strains Herts/33 and F48E8 significantly enhanced cell migration (Herts/33: 180 cells on average; F48E8: 242 cells on average) compared to the La Sota and mock groups (La Sota: 108 cells on average; Mock: 87 cells on average) at 6 hpi (Fig 1C). Interestingly, the migratory cell count for these virulent strains had notably decreased at the later stage of infection (24hpi, Herts/33: 36 cells on average; F48E8: 21 cells on average; La Sota: 124 cells on average; Mock: 94 cells on average) (S1 Fig). This observation could be attributed to the fact that compromised TJs in the early phase of infection facilitated cellular migration, whereas the extensive cell death occurring in the later stages of infection leads to a diminished migratory rate. Similar to the results of cellular pathology, the F48E8 strain demonstrated the most substantial induction of cell migration at the early stages of infection. TJs serve as the primary structural components responsible for forming the intercellular barrier. To evaluate the impact of NDV on lung epithelial barrier permeability, an *in vitro* model was constructed using A549 cells. TEER readings reflect the permeability barrier formed by tight junctions in cultured cells. A decrease in TEER values points to possible disruption of these cell-cell connections [34]. Here, TEER values showed a significant decrease from 6 to 24 hpi following infection exclusively with the virulent NDV, particularly the F48E8 strain, when compared to controls, indicating that the virulent NDV infection exacerbates barrier injury (Fig 1D). Therefore, these findings suggest that NDV infection can disrupt the intercellular TJ integrity.

To confirm the direct impact of NDV infection on TJs, we assessed expression levels of TJ-associated proteins. Here, we observed that both virulent NDV strains significantly reduced the protein expression levels of ZO-1 and OCLN in A549 cells, whereas no appreciable changes were detected in the La Sota-vaccinated and mock-infected control groups. Notably, this downregulation of ZO-1 and OCLN by virulent NDV strains exhibited both time- and dose-dependent characteristics, with the F48E8 strain demonstrating a more pronounced effect compared to the other virulent strain (Fig 1E and 1F). We further evaluated the broad-spectrum regulatory effects of NDV on TJ proteins across multiple cell lines (HeLa, HD11, CTE) by establishing infection models. Consistent with findings in A549 cells, only the virulent F48E8 strain induced significant cellular damage, whereas avirulent NDV La Sota and controls did not (S2A–S2C Fig). Western blot assays revealed stable expression of ZO-1 and OCLN in all cell lines tested. Upon infection, virulent F48E8 markedly reduced TJ protein levels in HD11 and CTE cells compared to La Sota and controls. Notably, NDV exhibited a significantly weaker

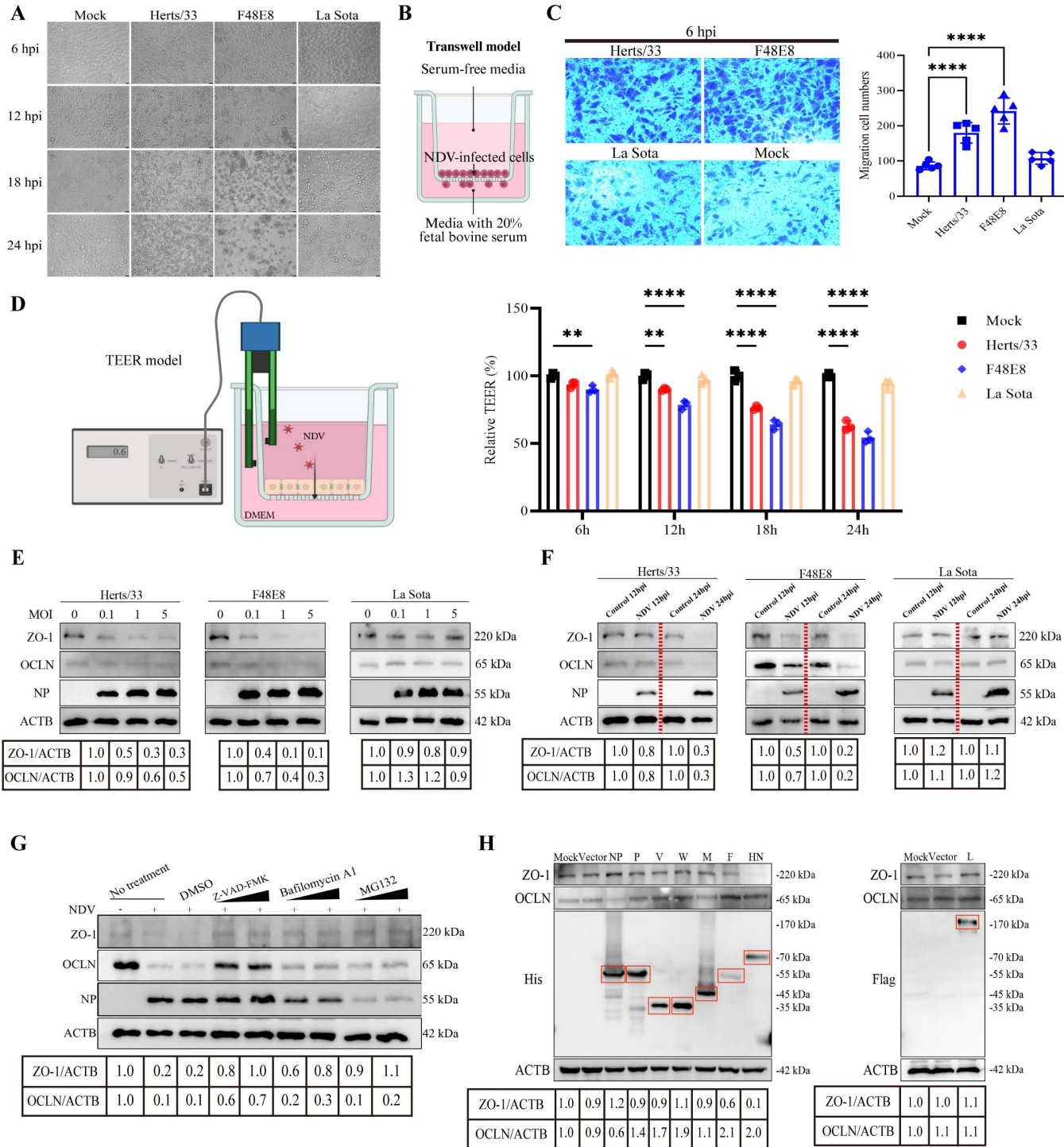

**Fig 1. NDV disrupts the TJ integrity and diminishes the expression levels of TJ-associated proteins. (A)** The cellular morphology was monitored in NDV- and mock-infected A549 cells at an MOI of 1 from 6 to 24 hpi under light microscopy. Scale bar: 20 μm. **(B)** Transwell assay was employed to determine the cell migration. The transwell model was established that the cells in serum-free medium were introduced into the upper chamber, and the lower chamber was then supplemented with complete culture medium supplemented with 20% fetal bovine serum to serve as a migratory stimulus. Created in BioRender. Lu, D. (2025) https://BioRender.com/hx6j612. **(C)** Transwell assay was employed to determine the cell migration in NDV- and mock-infected A549 cells at an MOI of 0.1 at 6 hpi. A suspension containing 4 × 10⁴ cells was utilized for the transwell assay. Scale bar: 50 μm. The

quantitative assessment of the migratory cell population. The infected groups were compared to the mock group, and statistical analysis was carried out. **(D)** The detection of barrier permeability following NDV infection. *In vitro* model of the lung epithelial barrier infected with NDVs. NDVs infected the lung epithelial barrier for 6 h, 12 h, 18 h, and 24 h, and the TEER values were determined. Created in BioRender. Lu, D. (2025) https://BioRender.com/o7k6m61. **(E, F)** The expression levels of TJ-associated proteins OCLN and ZO-1 were assessed by western blotting across a range of NDV doses (0, 0.1, 1, and 5 MOI at 24 hpi) and at various time points post-infection (1 MOI at 12 and 24 hpi). The gray value of each protein was quantified by Image J and normalized to ACTB. The gray values of 0 MOI group and control group were considered as "1", respectively. **(G)** A549 cells were infected with NDV at 1 MOI and then treated with Z-VAD-FMK (20 μM), Bafilomycin A1 (1 μM), MG132 (20 μM). At 24 hpi, the cells were harvested for western blotting to detect OCLN and ZO-1 protein expression. The gray value of each protein was quantified by Image J and normalized to ACTB. The gray value of control group was considered as "1". **(H)** A549 cells were transfected with eukaryotic expression plasmids encoding each viral protein. At 36 hpt, the expression levels of OCLN, ZO-1, and the viral proteins were assessed. The gray value of each protein was quantified by Image J and normalized to ACTB. The gray values of mock group were considered as "1". Red boxes indicate the positions of the respective viral proteins in the electrophoresis gel. ** $P < 0.01$; **** $P < 0.0001$.

ability to downregulate ZO-1 and OCLN in HeLa cells (S2D–S2F Fig). CLDN1 is a classic TJ protein, frequently used alongside ZO-1 and OCLN as markers in TJ research. In this study, we detected CLDN1 expression in these cells and found it was only present in A549 and CTE cells. Interestingly, CLDN1 was undetectable in both HeLa and HD11 cells. This finding aligns with previous reports confirming the absence of CLDN1 in HeLa cells [35], and notably, our study provides the first evidence that HD11 cells also lack CLDN1 expression. Furthermore, our results indicated that virulent NDV significantly downregulated CLDN1 expression in A549 and CTE cells, whereas avirulent NDV did not exert a significant effect (S3 Fig). Based on these results, we selected stably expressed ZO-1 and OCLN, and stable cell lines A549 and HD11 (from different species), for further study. Intriguingly, qPCR analysis showed that NDV infection did not decrease the mRNA levels of these TJ genes, and infection with virulent strains even upregulated them (S4 Fig). These findings suggest that NDV-mediated suppression of TJ protein expression operates at the translational level, not the transcriptional level. Caspase-hydrolase, autophagy-lysosome, and ubiquitin-proteasome constitute the principal protein degradation pathways [36]. To investigate the pathway by which NDV degrades TJ proteins, A549 cells were infected with the virulent NDV and subsequently treated with Z-VAD-FMK (a caspase inhibitor), Bafilomycin A1 (an autophagy inhibitor), and MG132 (a ubiquitin-proteasome inhibitor). Western blot analysis revealed that treatment with all three inhibitors—Z-VAD-FMK, Bafilomycin A1, and MG132—led to a restoration of ZO-1 protein levels; however, only Z-VAD-FMK treatment could significantly restore the expression of OCLN. These findings suggest that infection of the virulent NDV can modulate the degradation of TJ protein via diverse pathways, with the caspase-hydrolase pathway exerting a predominant influence (Fig 1G). To assess the impact of NDV viral proteins on TJs, we examined the expression levels of TJ proteins, ZO-1 and OCLN, following transfection. We initially constructed plasmids expressing eight NDV proteins: six structural (NP, P, M, F, HN, L) and two non-structural (V, W). Western Blot analysis confirmed successful transfection and expression of all viral proteins. Notably, distinct viral proteins exerted differential effects on ZO-1 and OCLN. Specifically, NP protein significantly downregulated OCLN expression, whereas F and HN proteins significantly reduced ZO-1 levels. These findings suggest that NDV's regulation of TJs is not mediated by a single protein but involves a complex, synergistic interplay among multiple viral proteins (Fig 1H). Therefore, virulent NDV disrupts TJs by targeting OCLN and ZO-1, a process involving complex degradation pathways and interactions between viral proteins.

## Reduced expression of TJ proteins facilitates both viral replication and spread of NDV

NDV reduces the expression of TJ proteins, a process previously implicated in the replication cycles of diverse viruses [37,38]. To confirm whether the TJ proteins exerted a regulatory influence on viral replication, we next assessed the effect of TJ protein knockdown on NDV replication. Considering the substantial impact of the virulent F48E8 strain on the TJ homeostasis, we were poised to utilize F48E8 for subsequent investigation. We employed siRNA-OCLN and siRNA-ZO-1 to generate A549/OCLN-KD and A549/ZO-1-KD cells, respectively. Western blot assay confirmed significant knockdown of these TJ genes (Fig 2A and 2B). Concurrently, the CCK-8 assay was conducted to verify that the reduction of OCLN

and ZO-1 levels did not affect cell viability (Fig 2C and 2D). To examine the impact of OCLN and ZO-1 on viral replication, A549/OCLN-KD and A549/ZO-1-KD cells were infected with NDV. The viral load in cells and supernatants was quantified

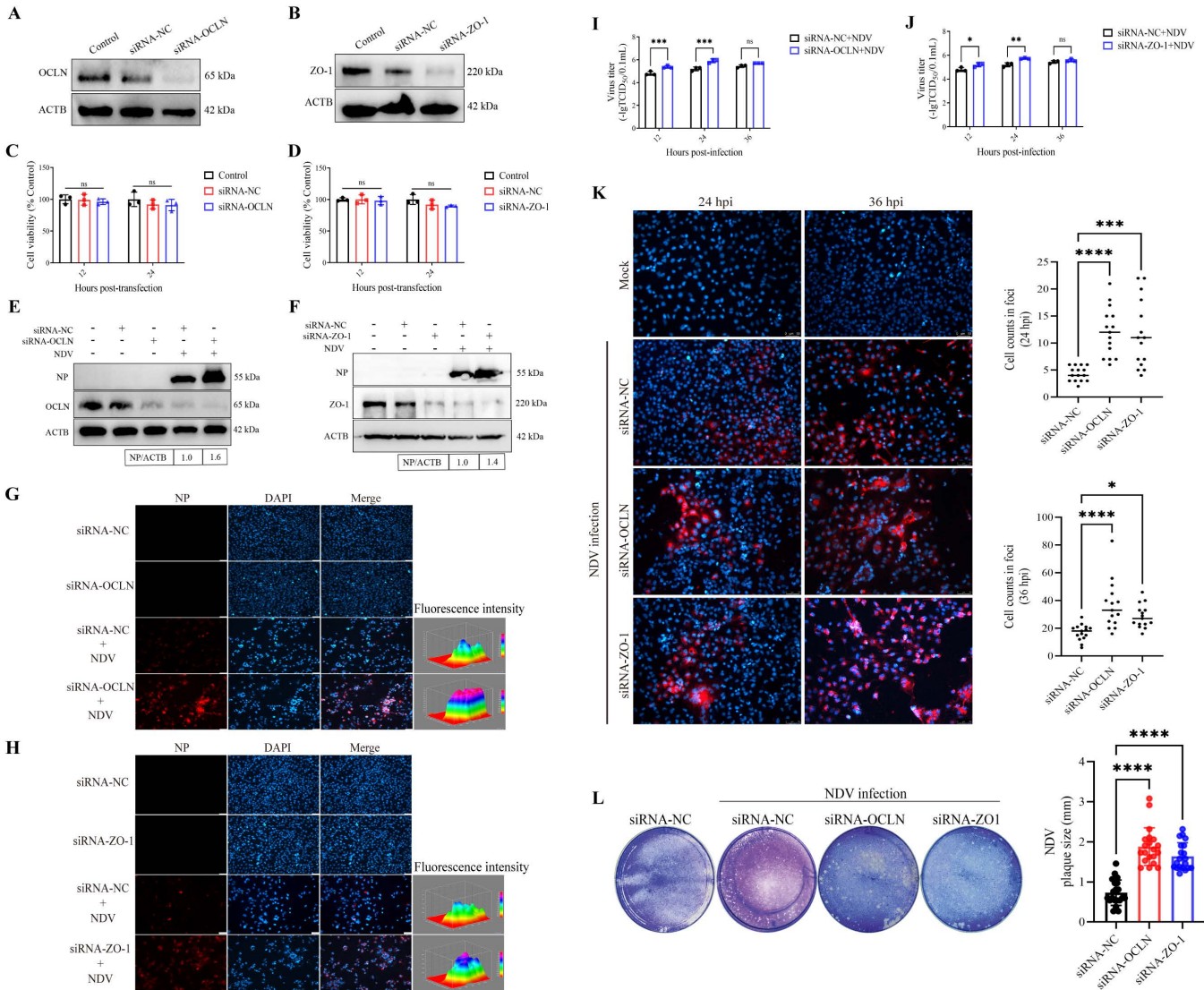

**Fig 2. Downregulation of TJ proteins facilitates viral replication and spread.** siRNA-OCLN and siRNA-ZO-1 (60 pmol) were employed to generate A549/OCLN-KD and A549/ZO-1-KD cells, respectively. siRNA-NC transfected cells were recognized as the control A549/ctrl. **(A, B)** Western blot assay determined the protein levels of OCLN and ZO-1 following siRNA transfection. **(C, D)** CCK-8 assay determined the cell viability following siRNA transfection. The transfected groups were compared to the control group, and statistical analysis was carried out. **(E, F)** A549/ctrl, A549/OCLN-KD and A549/ZO-1-KD cells were infected with NDV at 1 MOI for 24 h. The viral replication in cells was evaluated by western blotting. The gray value of each protein was quantified by Image J and normalized to ACTB. The gray value of NDV-infected A549/ctrl group was considered as "1". **(G, H)** A549/ctrl, A549/OCLN-KD and A549/ZO-1-KD cells were infected with NDV at 0.1 MOI for 24 h. The viral replication in cells was evaluated by IFA. Fluorescent intensities were quantified using Image J. Scale bar: 50 μm. **(I, J)** A549/ctrl, A549/OCLN-KD and A549/ZO-1-KD cells were infected with NDV at 0.1 MOI for 12, 24, and 36 h. The viral load in supernatants was detected by $TCID_{50}$. **(K)** A549/ctrl, A549/OCLN-KD and A549/ZO-1-KD cells were infected with NDV at a low dose (MOI = 0.00001) for 24 and 36 h, and the foci in cells were evaluated by IFA. Fifteen random lesions per treatment group were chosen, and the extent of viral dissemination was determined by counting the number of cells within each lesion. Scale bar: 50 μm. **(L)** A549/ctrl, A549/OCLN-KD and A549/ZO-1-KD cells were infected with NDV at a low dose (MOI = 0.00001) within the agarose gel for 60 h. The relative plaque size was quantified using ImageJ software. * $P < 0.05$; ** $P < 0.01$; *** $P < 0.001$; **** $P < 0.0001$; ns, no significant difference.

using western blot, IFA, and $TCID_{50}$ assays, respectively. Western blot (Fig 2E and 2F) and IFA (Fig 2G and 2H) assays collectively showed higher viral NP protein in OCLN or ZO-1 knock-down infected cells than in control infected cells. $TCID_{50}$ assays revealed that viral titers in the supernatant of cells with OCLN or ZO-1 knockdown were markedly elevated compared to the control infection group at 12 and 24 hpi. However, no significant difference in titer was observed at 36 hpi, potentially due to the saturation of viral replication during the late stage of infection (Fig 2I and 2J). Macrophages are one of the susceptible target cells for NDV, and we also the impact of TJ proteins on viral replication in avian HD11 cells. We employed siRNA-OCLN and siRNA-ZO-1 to generate HD11/OCLN-KD and HD11/ZO-1-KD cells, respectively. Western blot confirmed substantial knockdown of TJ genes, and the CCK-8 assay confirmed that decreased OCLN and ZO-1 levels did not impact cell viability (S5A and S5B Fig). To examine the impact of OCLN and ZO-1 on viral replication, HD11/OCLN-KD and HD11/ZO-1-KD cells were infected with NDV. The viral load in cells and supernatants was quantified using western blot and $TCID_{50}$ assays, respectively. Western blot and $TCID_{50}$ assays indicated significantly enhanced viral replication in NDV-infected HD11/OCLN-KD and HD11/ZO-1-KD cells versus HD11/ctrl cells (S5C and S5D Fig).

TJ proteins have been demonstrated to be involved in the spread of virus via cell-to-cell contact [39]. Here, we hypothesized that TJ proteins enhanced NDV replication by facilitating its spread. To verify this hypothesis, we observed foci and plaques forming separately after NDV infection at a low MOI. The extent of cellular foci formation serves as an important indicator of viral spread leading to cytopathic effect, as observed with various viruses including hazara orthonairovirus (HAZV) [30], human parainfluenza virus (HPV) [31], etc. Foci formation results showed that NDV-infected A549/ctrl cells were observed as foci consisting of multiple cells at 24 hpi, and they subsequently expanded at 36 hpi. By contrast, viral foci were observed in A549/OCLN-KD and A549/ZO-1-KD cells significantly at 24 and 36 hpi. The size of viral foci was greater than those observed in A549/ctrl cells. The cell numbers within the viral foci observed in these cells were counted, when the remarkable difference was seen at 24 and 36 hpi. By 24 hpi, cell counts within viral foci in A549/OCLN-KD and A549/ZO-1-KD cells (each averaging 12 cells) was markedly elevated over A549/ctrl cells (averaging 4 cells). At 36 hpi, the viral foci in A549/OCLN-KD (averaging 37 cells) and A549/ZO-1-KD cells (averaging 29 cells) contained significantly more cells than in A549/ctrl cells (averaging 17 cells) (Fig 2K). Furthermore, the impacts of OCLN and ZO-1 on NDV spread between cells were measured by plaque assay and revealed that NDV infection in A549/OCLN-KD (averaging 1.88 mm) and A549/ZO-1-KD (averaging 1.64 mm) cells resulted in big-sized plaques compared to those formed in A549/ctrl cells (averaging 0.73 mm) (Fig 2L). Additionally, we conducted confirmatory experiments on HD11 cells. Foci formation results showed that the size of viral foci in HD11/OCLN-KD or HD11/ZO-1-KD cells was greater than those observed in HD11/ctrl cells. Viral foci in HD11/OCLN-KD cells (averaging 65 cells) and HD11/ZO1-KD cells (averaging 57 cells) contained more cells than in HD11/ctrl cells (averaging 11 cells), suggesting increased viral spread (S5E and S5F Fig). Therefore, these findings indicate that TJ proteins exert an inhibitory effect on NDV replication and spread. In other words, NDV facilitates its own replication and spread by downregulating the expression of these proteins.

## NDV induces redistribution of TJ structure, accompanied by F-actin fiber rearrangement and cage formation

We observed that the expression levels of OCLN and ZO-1 were decreased in the NDV-infected cells. OCLN and ZO-1 are essential constituents of the cellular TJ framework [40]. Therefore, we hypothesized that NDV infection could disrupt TJ structure. To test this hypothesis, we firstly examined alternations in the junction integrity *in vivo* following NDV infection. We successfully established an *in vivo* NDV infection model in chickens (Fig 3A). In this model, chickens infected with NDV exhibited rapid viral shedding, especially those inoculated with the virulent strain (S5 Table). In addition, *in vivo* studies observed decreased OCLN and ZO-1 expression in chicken lungs post-infection with NDV (Fig 3B). H&E staining results demonstrated that the alveolar walls in the control chicken lung tissue exhibited tight junction and a mesh-like arrangement, appearing hyalinized. In contrast, the alveolar architecture was compromised, with a loss of characteristic morphology, disrupted junctions, extensive cell death, infiltration of inflammatory cells, and hemorrhage following NDV infection (Fig 3C). Observing extensive inflammatory cell infiltration on H&E staining, we subsequently quantified

inflammatory cytokine levels in response to NDV infection. The qPCR results revealed that NDV infection induced a significant upregulation of mRNA levels of inflammatory cytokines, particularly for IL-1β, IL-6, and TNF-α (S6 Fig). In contrast, the avirulent NDV strain failed to elicit significant alterations in the expression of TJ proteins nor did it induce appreciable lung lesions (S7A and S7B Fig). Thus, these results indicate that NDV induces TJ damage in chicken lungs, which represents an important factor contributing to lung injury.

To vividly visualize the TJ structure, we utilized TEM and confocal microscopy to clearly observe intercellular TJ structure. TJ complexes were easily detectable at the intercellular junctions of neighboring cells in mock-infected group, but in NDV-infected cells, many cell-cell junctions were lost, and increased gaps between neighboring membranes were observed (Fig 3D). The confocal microscopy results showed that both OCLN and ZO-1 were distributed in a typical linear pattern along the layer of cell-to-cell contact in mock-infected cells. However, cellular localization of OCLN and ZO-1 was

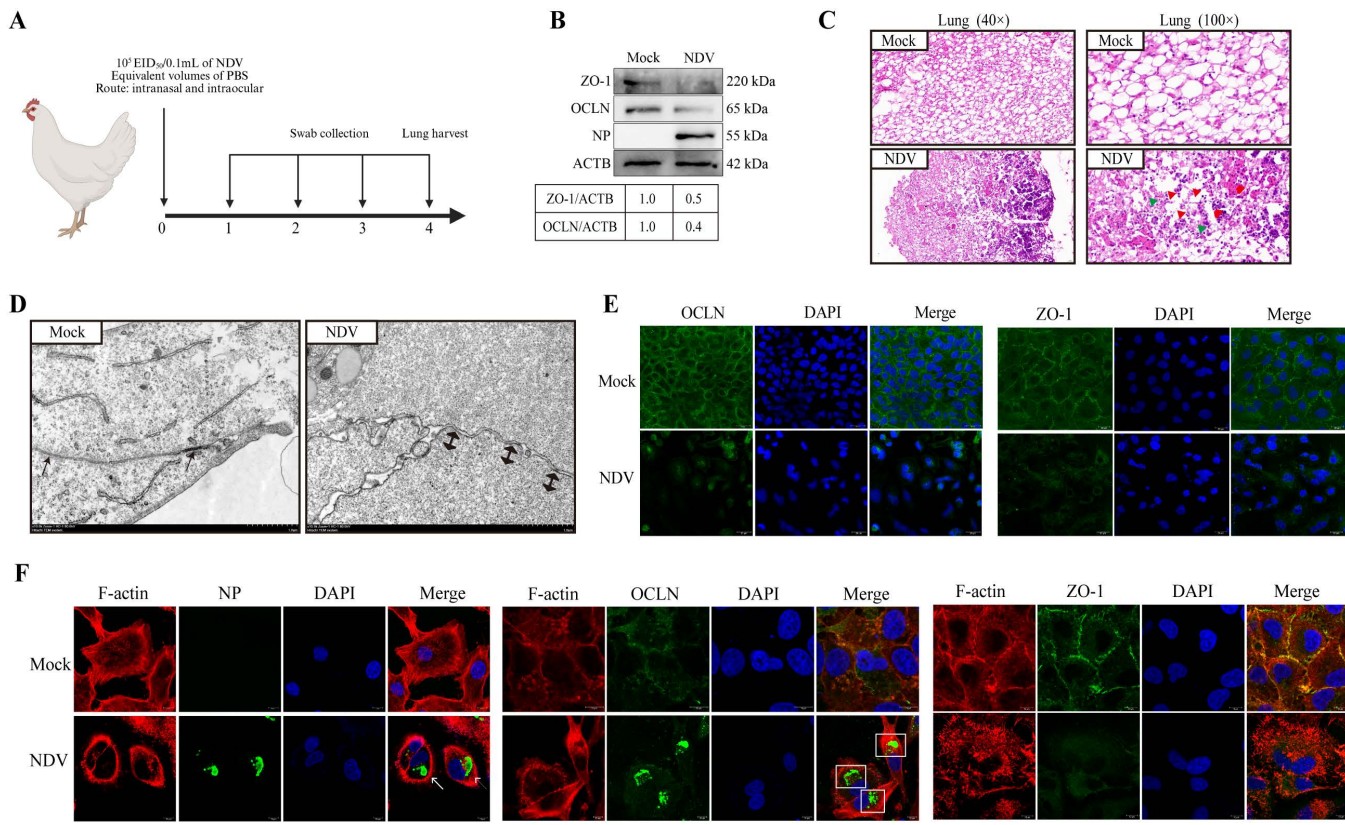

**Fig 3. NDV disrupts the TJ structure, accompanied by F-actin structural redistribution. (A)** 4-week-old SPF chickens were infected with $10^5$ EID$_{50}$ NDV via the intranasal and intraocular route for 4 days. PBS group was considered as the control group. The oropharyngeal and cloacal swabs were gathered daily from the infected chickens to evaluate virus shedding. The lung tissue was harvested to assess histopathological alterations and TJ-associated protein expression at 4 dpi. Created in BioRender. Lu, D. (2025) https://BioRender.com/ebond7q. **(B)** The OCLN and ZO-1 protein levels in lung tissue were detected by western blotting. The gray value of each protein was quantified by Image J and normalized to ACTB. The gray value of mock-infected group was considered as "1". **(C)** The pathological changes of lung tissue were observed by H&E staining (40× and 100×). The red triangle represents disrupted junctions, the green triangle represents necrotic cells, the red circle represents the infiltration of inflammatory cells, and the red box represents the hemorrhage. **(D)** The intercellular TJ structure was observed in NDV-infected and mock-infected A549 cells at 6 hpi by TEM assay. The solid black unidirectional arrow denotes a standard tight junction, whereas the solid black bidirectional arrow signifies an enlarged tight junction. Scale bar: 1 μm. **(E)** The localization of OCLN and ZO-1 was observed in NDV- (MOI = 0.1) and mock-infected A549 cells at 18 hpi by confocal microscopy. Scale bar: 20 μm. **(G)** The confocal microscopy was employed to assess the localization of F-actin and compared its distribution with that of OCLN and ZO-1 in NDV- (MOI = 0.1) and mock-infected A549 cells. The white arrow denotes the aggregation of NDV NP in the F-actin cage, whereas the dashed box highlights the aggregation of OCLN in the F-actin cage. Scale bar: 10 μm.

severely disrupted and a fragmented and clumpy pattern in NDV-infected cells, especially the phenomenon of clustering in OCLN (Fig 3E). F-actin can form TJ complexes with TJ proteins, which are crucial for the maintenance of cellular TJ homeostasis [7]. We further observed the localization of F-actin and compared its distribution with that of OCLN and ZO-1. In mock-infected cells, F-actin exhibited a typical fiber network structure throughout the cytoplasm. However, the F-actin fiber network was significantly disrupted, with evident fiber aggregation and cage-like formation observed. Notably, NP and OCLN aggregates were encased within F-actin cages, whereas fluorescence outside these cages was weak. This finding suggests their potential roles as hubs for viral replication and TJ redistribution. Furthermore, F-actin exhibited certain cellular co-localization with OCLN and ZO-1 in mock-infected cells, but the co-localization was compromised following NDV infection (Fig 3F). In contrast, the avirulent NDV strain failed to induce the obvious alternations of OCLN, ZO-1, and F-actin structure (S8A Fig). Hence, these findings suggest that NDV induces TJ redistribution, coupled with F-actin reorganization and cage formation.

## Vimentin serves as a critical molecular switch in the TJ injury during NDV infection

Vimentin structure serves as a critical mediator that promotes epithelial barrier dysfunction by modulating TJs, thereby playing a pivotal role in pathogen infection [32]. In addition, we have previously demonstrated that vimentin rearrangement contributes to NDV infection [27]. Here, we hypothesized that vimentin structure served as a pivotal regulatory factor in the TJ injury during NDV infection. To validate this hypothesis, we examined the structural changes of vimentin following NDV infection. Moreover, we employed non-cytotoxic IDPN to selectively inhibit vimentin rearrangement, subsequently evaluating the effect of weakened vimentin rearrangement on TJ homeostasis. The cytotoxicity of varying IDPN concentrations on A549 cells was evaluated using a CCK-8 assay. IDPN concentrations below 1.5% were found to be non-toxic (Fig 4A). In mock-infected and IDPN-treated cells, vimentin filaments formed a widespread, elongated network throughout the cytoplasm. Conversely, in virulent NDV-infected cells, these filaments appeared dense, shortened, and extensively accumulated around the nucleus. Notably, the virulent NDV induced vimentin cage formation, encapsulating NDV NP aggregates (Fig 4B). NP is a crucial NDV ribonucleoprotein (RNP) component, essential for viral genome synthesis along with P and L proteins [21]. Previous studies have confirmed that vimentin cages are sites for virus replication [41,42]. Here, the vimentin cage may also promote NDV replication by influencing NP function. In contrast, the avirulent NDV failed to induce significant alterations in vimentin structure, with its morphology resembling that of mock-infected cells (S8B Fig). Subsequently, confocal microscopy, western blotting and TCID$_{50}$ assays demonstrated that IDPN treatment remarkably impaired viral replication both within the cells and in the supernatant (Fig 4B–4D). Of note, confocal microscopy analysis revealed a significant association between the redistribution vimentin, F-actin, and OCLN following NDV infection, indicating a tight spatial correlation within these cellular structures. Briefly, NDV infection elicited the formation of a vimentin cage, which was found to be colocalized with an F-actin cage, as indicated by the white arrowheads. Echoing the phenomenon of colocalization observed between the F-actin cage and OCLN, the vimentin cage wrapped around the NDV-induced OCLN aggregations, as indicated by white dashed boxes. In contrast, IDPN treatment significantly inhibited NDV-induced vimentin rearrangement and cage formation, which was consistent with our previous results [27]. Additionally, IDPN treatment successfully inhibited the NDV-induced redistribution of ZO-1, OCLN, and F-actin, as compared to untreated NDV-infected cells. Vimentin/F-actin cage formation was also inhibited (Fig 4E). These findings indicate that vimentin functions as a critical molecular switch in mediating TJ injury induced by NDV.

To elucidate the impact of NDV replication activity on TJ proteins, we inactivated the virus using UV treatment and employed chicken embryos to confirm the lack of replication in the inactivated virus. After inactivation, we analyzed the expression levels of OCLN and ZO-1 proteins through western blotting. Subsequently, we employed confocal microscopy to observe the structure of OCLN, ZO-1, F-actin, and vimentin within the cells. Western blot analysis revealed that UV-inactivated NDV did not diminish the expression levels of OCLN and ZO-1 proteins (S9A Fig). Moreover, confocal microscopy observations indicated that UV-inactivated NDV failed to induce structural rearrangements of OCLN, ZO-1, F-actin,

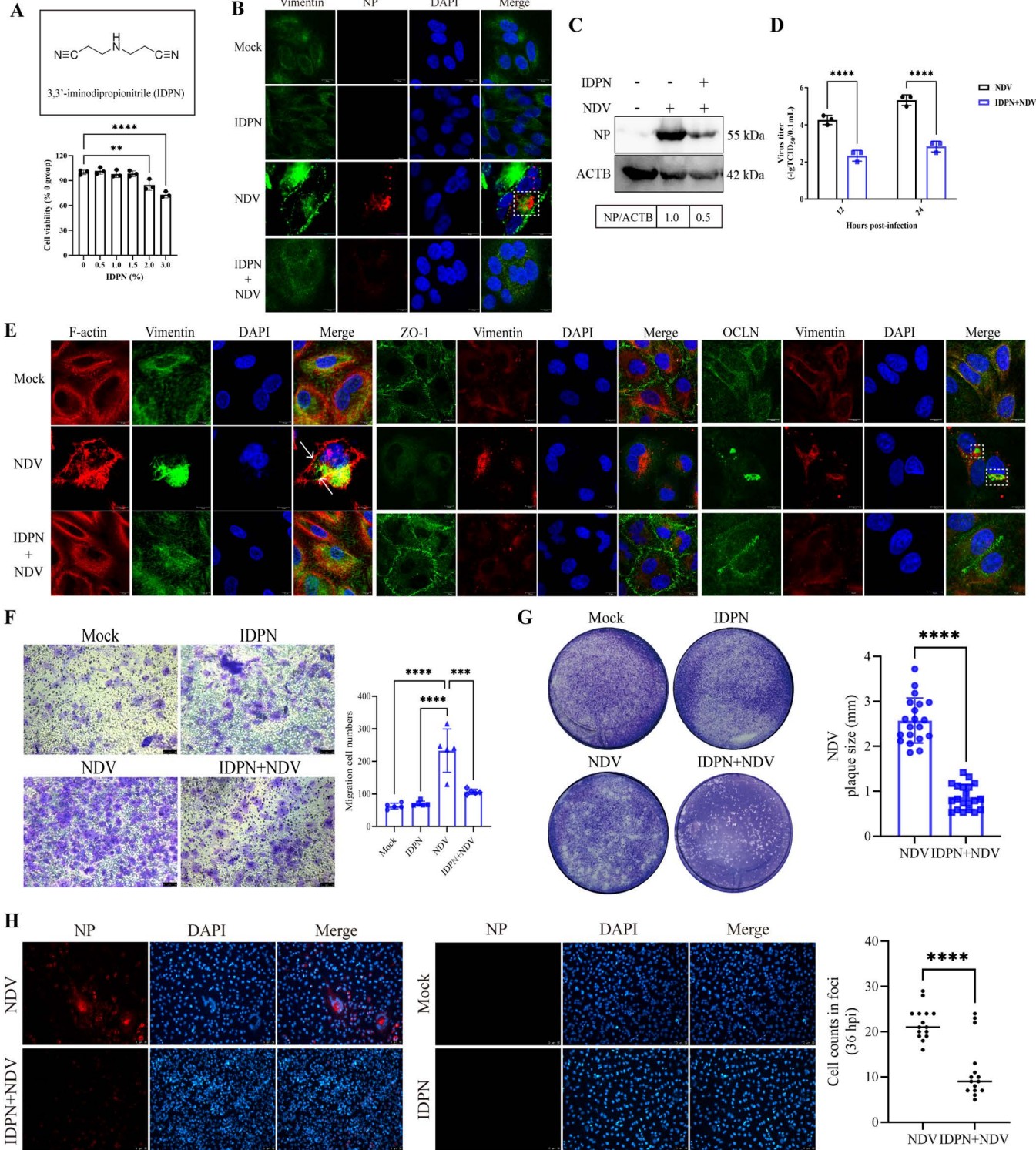

**Fig 4. NDV-induced vimentin rearrangement contributes to TJ injury and facilitates viral replication and spread. (A)** Chemical formula of the IDPN pharmaceutical compound. Cytotoxicity of IDPN was assessed using the CCK8 assay. **(B)** Structural examination of vimentin in A549 cells infected with NDV at 0.1 MOI was conducted, with a comparative analysis of the fibrous structures in the presence and absence of IDPN (1.5%). The dashed box highlights the aggregation of NDV NP in the vimentin cage. Scale bar: 10 μm. **(C)** The replication levels of NDV (MOI = 0.1) were quantified by western blotting at 24 hpi, comparing NP levels in the presence and absence of IDPN (1.5%). **(D)** The replication levels of NDV (MOI = 0.1) were quantified

using the TCID$_{50}$ assay at 12 and 24 hpi, comparing the viral titers in the presence and absence of IDPN (1.5%). **(E)** The co-localization of vimentin with F-actin, ZO-1, and OCLN was monitored by confocal microscopy in NDV- (MOI = 0.1) and mock-infected A549 cells at 18 hpi, which was compared in the presence and absence of IDPN (1.5%). The white arrow indicates the rearranged cage-like structure of vimentin and F-actin, whereas the dashed box highlights the aggregation of OCLN in the vimentin cage. Scale bar: 10 μm. **(F)** The cell migration by NDV (MOI = 0.1) were assessed at 6 hpi using the transwell assay, comparing the amounts of migrated cells in the presence and absence of IDPN (1.5%). A suspension containing $4 \times 10^4$ cells was utilized for the transwell assay. Scale bar: 50 μm. **(G)** The plaque formation was assessed in NDV-infected A549 cells (MOI = 0.00001) using the plaque assay, comparing the plaque size in the presence and absence of IDPN (1.5%). The relative plaque size was quantified using ImageJ software. **(H)** The foci formation was measured in NDV-infected A549 cells (MOI = 0.00001) by IFA, comparing the number of cells within foci in the presence and absence of IDPN (1.5%). Scale bar: 50 μm. ** $P < 0.01$; *** $P < 0.001$; **** $P < 0.0001$.

and vimentin, resembling the structure observed in mock-infected cells (S9B Fig). Thus, the effective viral replication is required for the modulation of TJ homeostasis.

### Vimentin rearrangement is necessary for TJ injury-mediated enhancement of viral spread and inflammation

Our previous work showed TJ integrity affected virus spread and vimentin rearrangement impacted TJ integrity. This study extended that to examine vimentin's role in NDV spread. Here, we performed a transwell assay to determine the impact of vimentin rearrangement on cell migration induced by NDV. The results showed that NDV triggered massive cell migration (averaging 233 cells), while IDPN treatment significantly inhibited this characteristic of NDV (averaging 107 cells) (Fig 4F). The result indicated that vimentin rearrangement was required for cell migration by NDV. Subsequently, we further confirmed the effect of vimentin rearrangement on NDV spread by measuring plaque and lesion formation. The plaque analysis showed that the plaque size induced by NDV in IDPN-treated cells (averaging 0.87 mm) was significantly smaller than that of the untreated infection group (averaging 2.57 mm) (Fig 4G). Furthermore, lesion formation results showed that NDV-infected cells were observed as massive foci consisting of multiple cells, whereas IDPN treatment impaired the foci formation. The cell numbers within the viral foci observed in these cells were counted, when the remarkable difference was seen between the normal infection group and IDPN-treated infection group. The cell number within the viral foci in the normal infection group (averaging 22 cells) was significantly more than those in IDPN-treated infection group (averaging 11 cells) (Fig 4H).

To elucidate whether the vimentin rearrangement can directly influence the TJ structure, we further used ajoene, a vimentin rearrangement activator, for subsequent analysis. Ajoene, a phytochemical derived from garlic, induces rearrangement and condensation of the vimentin filament network by binding to vimentin at Cys-328, ultimately leading to its disruption [43]. The CCK8 assay showed that ajoene concentrations below 10 μM were found to be non-toxic (Fig 5A). Confocal microscopy confirmed ajoene as an effective inducer of vimentin rearrangement under physiological conditions (Fig 5B). Notably, ajoene-induced vimentin rearrangement led to changes in the distribution of F-actin, ZO-1, and OCLN (indicated by white arrows), suggesting that vimentin structure was vital for maintaining TJ structural integrity (Fig 5B–5D). This finding verified that the TJ structure could be directly governed by the vimentin rearrangement. Moreover, we evaluated the effects of ajoene-induced vimentin rearrangement on viral replication and spread. Western blot and TCID$_{50}$ analyses demonstrated enhanced replication of NDV in cells treated with ajoene (Fig 5E and 5F). The transwell assay revealed that NDV induced cell migration (averaging 118 cells), and ajoene treatment significantly potentiated NDV-induced migration (averaging 215 cells) (Fig 5G). IFA results showed NDV-infected cells formed multi-cellular foci, which were augmented by ajoene treatment. Counting the cells within viral foci revealed a significant difference between the control infection and ajoene-treated groups. The ajoene-treated group had a notably higher cell count within viral foci (averaging 41 cells) compared to the control group (averaging 16 cells) (Fig 5H). Additionally, we measured cytokine levels after IDPN or ajoene treatments. The qPCR results showed that IDPN treatment inhibited NDV-induced inflammatory cytokine activation, whereas ajoene treatment markedly elevated cytokine expression levels (S10 Fig).

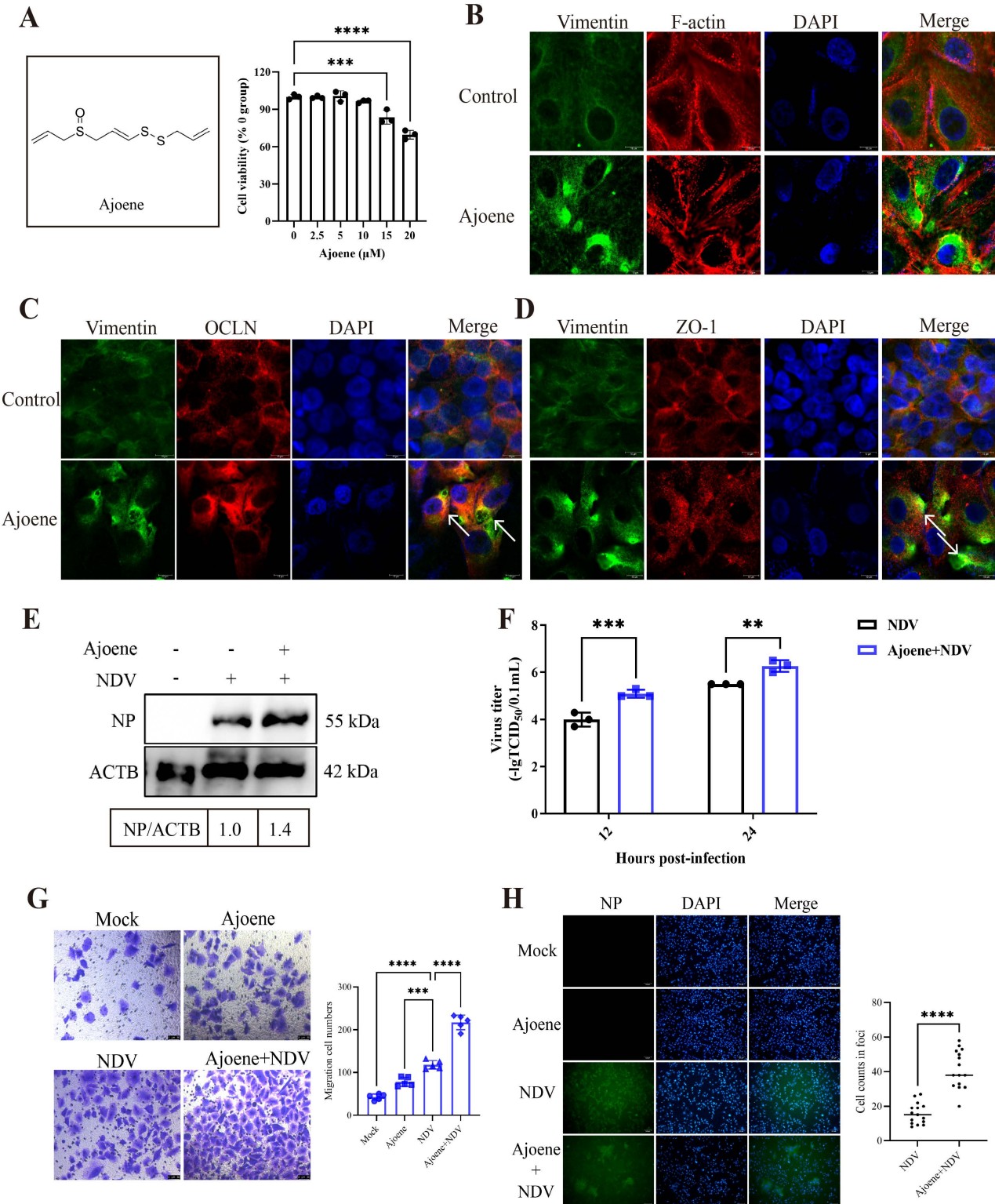

**Fig 5. Activated vimentin rearrangement directly drives TJ injury, facilitating viral replication and spread. (A)** Chemical formula of the ajoene pharmaceutical compound. Cytotoxicity of ajoene was assessed using the CCK8 assay. **(B)** Structural examination of vimentin and F-actin in A549 cells treated with ajoene (10 μM) was conducted. Scale bar: 10 μm. **(C)** Structural examination of vimentin and OCLN in A549 cells treated with ajoene (10 μM) was conducted. The white arrow indicates the rearranged vimentin and OCLN. Scale bar: 10 μm. **(D)** Structural examination of vimentin and ZO-1

in A549 cells treated with ajoene (10 µM) was conducted. The white arrow indicates the rearranged vimentin and ZO-1. Scale bar: 10 µm. **(E)** The NP protein levels were examined in NDV-infected (MOI = 1) A549 cells treated with or without ajoene (10 µM) by western blotting. The gray value of each protein was quantified by Image J and normalized to ACTB. The gray value of NDV-infected untreated group was considered as "1". **(F)** The replication levels of NDV (MOI = 0.1) were quantified using the TCID$_{50}$ assay in A549 cells, comparing the viral titers in the presence and absence of ajoene (10 µM). **(G)** The cell migration by NDV (MOI = 0.1) were assessed at 6 hpi using the transwell assay in A549 cells, comparing the amount of migrated cells in the presence and absence of ajoene (10 µM). A suspension containing $2 \times 10^4$ cells was utilized for the transwell assay. Scale bar: 50 µm. **(H)** The cell foci were measured in NDV-infected A549 cells (MOI = 0.00001) by IFA, comparing the number of cells within foci in the presence and absence of ajoene (10 µM). Scale bar: 100 µm. ** $P < 0.001$; *** $P < 0.001$; **** $P < 0.0001$.

Furthermore, we confirmed that vimentin rearrangement was essential for increased viral spread and inflammation in avian cells due to TJ injury. As stable avian epithelial cells were unavailable for our study, we evaluated TJ-like structure formation in the avian macrophage cell line HD11, which is susceptible to NDV. Using confocal microscopy, we examined the localization of TJ proteins in mock-infected HD11 cells and observed that both OCLN and ZO-1 proteins formed distinct fluorescent structures at cell contacts, arranged in a honeycomb-like pattern, indicative of a typical TJ-like structure (S11 Fig). This finding aligns with previous reports documenting TJ-like structures in certain non-epithelial cells [8,9,44]. Given that HD11 cells stably express TJ proteins and form these TJ-like structures, we considered them a relevant avian cell model for our subsequent validation studies. We utilized confocal microscopy to investigate the structural changes in OCLN, ZO-1, cytoskeletal vimentin, and F-actin following NDV infection. In NDV-infected cells, these proteins showed severe alterations, appearing fragmented and clumped, similar to changes observed in A549 cells. OCLN, vimentin, and F-actin accumulated significantly around the nucleus, forming cage-like structures (marked by white arrowheads), while the ZO-1 structure appeared degraded. Notably, NDV caused vimentin and F-actin to form cages around OCLN and NDV NP aggregations (S13A Fig). We subsequently tested vimentin's role in TJ injury in HD11 cells by assessing NDV-induced vimentin rearrangement. IDPN significantly inhibited NDV-induced rearrangement while demonstrating no non-toxic to cells at concentrations ≤1.5% and no influence on normal vimentin structure (S12A and S12B Fig). In contrast, IDPN treatment successfully prevented these NDV-induced redistributions (S13A Fig). IDPN significantly reduced NDV replication in cells and supernatant, as shown by western blotting and TCID$_{50}$ assays (S13B and S13C Fig). NDV promoted cell migration (average 153 cells), which IDPN markedly inhibited (average 102 cells) (S13D Fig). NDV caused infected cells to form large foci, but IDPN treatment impaired foci formation. Cells within foci were significantly fewer in the IDPN-treated group (average 9 cells) compared to the NDV-only group (average 38 cells) (S13E Fig). These results demonstrate the role of vimentin in NDV spread within HD11 cells. Additionally, we used ajoene to examine vimentin rearrangement's direct effects on TJs and viral infection in HD11 cells. Confocal microscopy confirmed ajoene induced vimentin rearrangement without significant toxicity below 10 µM (S14A Fig). The rearrangement coincided with F-actin, OCLN, and ZO-1 clustering, showing vimentin directly influences the TJ-like structure in HD11 cells. We also assessed ajoene's effect via vimentin rearrangement. This led to increased NDV replication (S14B and S14C Fig), enhanced cell migration (Ajoene-NDV: 217 cells vs NDV: 118 cells) (S14D Fig), and larger viral foci (Ajoene-NDV: 86 cells vs NDV: 33 cells) (S14E Fig). Consistent with A549 findings, ajoene boosted inflammation, whereas IDPN reduced it (S15 Fig). These findings indicate that vimentin rearrangement promotes viral replication and spread through the induction of TJ injury.

### MLC/p-MLC activation facilitates vimentin-mediated TJ injury, thereby promoting NDV replication and spread

MLC/p-MLC activation is known to induce actomyosin contraction, which can lead to TJ disruption [5]. Here, we examined MLC phosphorylation in A549 cells following NDV infection. Cells were either mock infected or infected with NDV at 1 MOI. Western blot analysis revealed increased p-MLC levels in NDV-infected cells compared to mock-infected controls (Fig 6A), indicating that NDV infection indeed induces MLC phosphorylation. To assess the functional significance of MLC/p-MLC activation in NDV-induced TJ injury and viral infection, we employed the MLC inhibitor blebbistatin. Prior to infection, cells were treated with blebbistatin. Concentrations below 20 µM were determined to be noncytotoxic via CCK8

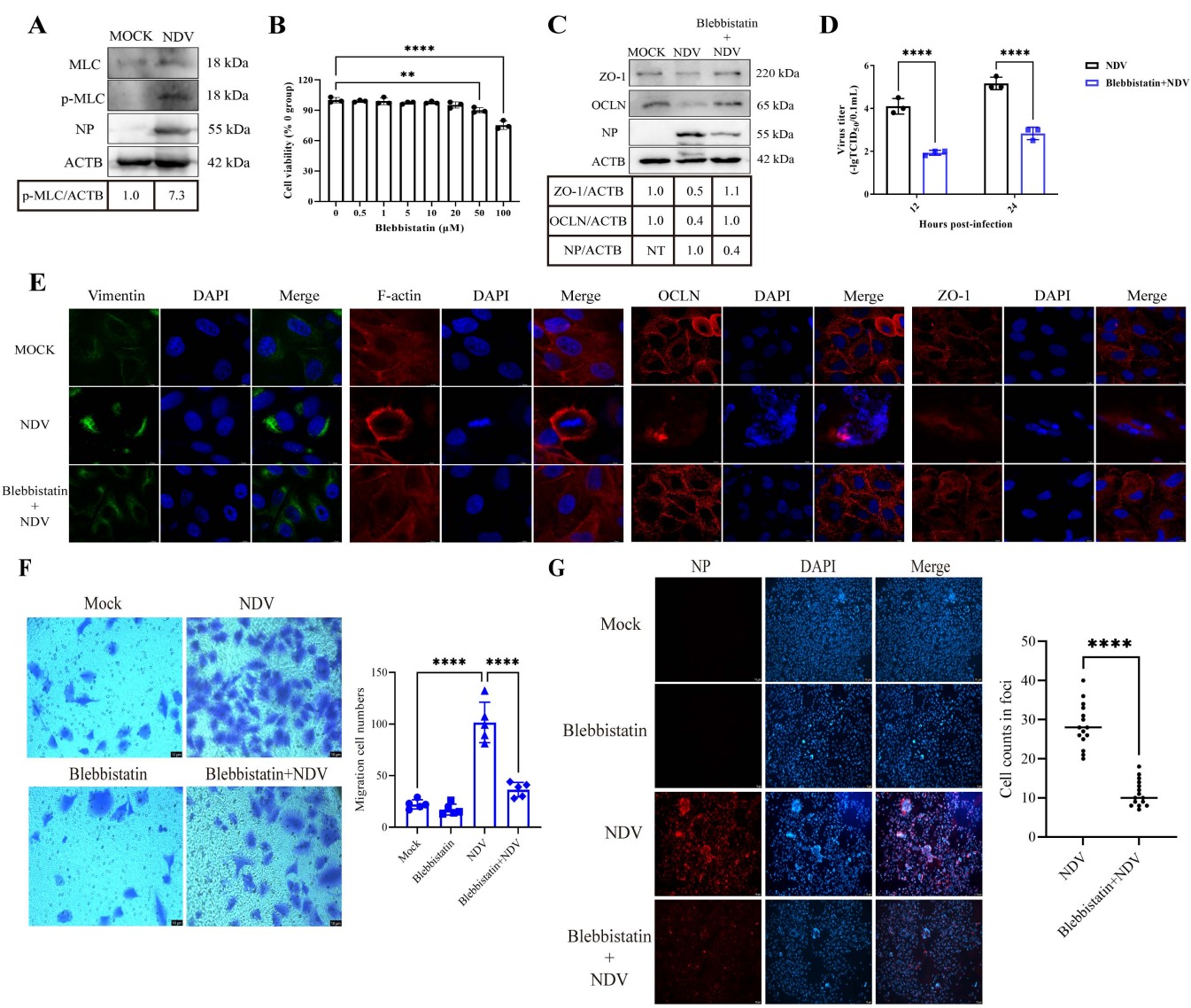

**Fig 6. MLC/p-MLC activation drives vimentin-mediated TJ injury, boosting NDV replication and spread. (A)** A549 cells were mock infected or NDV-infected at 1 MOI for 24 h, and the MLC and p-MLC protein levels were assessed by western blotting. The gray value of each protein was quantified by Image J and normalized to ACTB. The gray value of mock-infected group was considered as "1". **(B)** Cytotoxicity of the MLC inhibitor blebbistatin was assessed using the CCK8 assay. **(C)** The OCLN, ZO-1, and NP protein levels were examined in NDV-infected (MOI = 1) A549 cells treated with or without blebbistatin (20 μM) by western blotting. The gray value of each protein was quantified by Image J and normalized to ACTB. Gray values for OCLN and ZO-1 in the mock-infected group were normalized to "1", and NP values in the NDV-infected untreated group were similarly normalized to "1". NT indicates not test. **(D)** The replication levels of NDV (MOI = 0.1) were quantified using the $TCID_{50}$ assay in A549 cells, comparing the viral titers in the presence and absence of blebbistatin (20 μM). **(E)** Structural analysis of vimentin, F-actin, OCLN, and ZO-1 in NDV-infected (0.1 MOI) A549 cells (treated with or without blebbistatin, 20 μM) was performed, with mock-infected cells serving as a control group. Scale bar: 10 μm. **(F)** The cell migration by NDV (MOI = 0.1) were assessed at 6 hpi using the transwell assay in A549 cells. Migration levels were compared between blebbistatin (20 μM)-treated and untreated cells, with mock-infected cells as controls. A suspension containing $2 \times 10^4$ cells was utilized for the transwell assay. Scale bar: 10 μm. **(G)** The cell foci were measured in NDV-infected A549 cells (MOI = 0.00001) by IFA. The number of cells within foci was compared between blebbistatin (20 μM)-treated and untreated cells. Scale bar: 10 μm. ** $P < 0.01$; **** $P < 0.0001$.

assay (Fig 6B). Blebbistatin treatment significantly mitigated NDV-induced downregulation of TJ proteins levels and viral replication (Fig 6C and 6D). We next investigated the effect of MLC/p-MLC activation on the TJ complex structure using confocal microscopy. We examined the localization of vimentin, F-actin, OCLN, and ZO-1. Compared to NDV-infected cells, blebbistatin treatment prevented the NDV-induced redistribution of these cytoskeletal and TJ proteins, causing their localization patterns to more closely resemble those of mock-infected cells (Fig 6E). This further supports MLC/p-MLC involvement in TJ rearrangement upon NDV infection. Given MLC/p-MLC's role in TJ redistribution and its impact on viral replication, we next explored its contribution to NDV spread. Both transwell migration and foci formation assays demonstrated that MLC/p-MLC activation promotes NDV dissemination. Specifically, blebbistatin treatment significantly inhibited the migration of NDV-infected cells (Blebbistatin-NDV: 36 cells vs NDV: 102 cells) and the formation of viral foci (Blebbistatin-NDV: 11 cells vs NDV: 28 cells) (Fig 6F and 6G). Collectively, these findings suggest that MLC/p-MLC activation facilitates NDV-induced TJ injury, thereby promoting viral replication and spread.

## Discussion

TJs are ubiquitously present across diverse tissues and cellular compartments in the body, where they fulfill a pivotal function in establishing tissue barriers and preserving cellular homeostasis [45]. As the primary stabilizers of the epithelial barrier, TJs function as the frontline physical defense against pathogens [2]. Conversely, infection with various pathogens can disrupt TJs, which in turn facilitates the establishment of infection [39,46]. As a significant global avian pathogen, NDV spreads extensively among both domestic poultry and wild birds, posing a serious threat to poultry production worldwide [47]. Its pathogenic mechanisms have consistently been a popular area of research interest. Despite extensive research elucidating many aspects of NDV pathogenesis, the precise effects of its infection on host TJ integrity and function, and the underlying mechanisms of this interaction, are still unclear. Therefore, our study aims to elucidate the interaction mechanism between NDV and TJs, employing both *in vitro* and *in vivo* experiments for this purpose.

NDV, much like avian influenza virus, can infect hosts via the respiratory tract and subsequently spread rapidly [48]. The A549 cell line, derived from lung epithelium, possesses characteristics of both respiratory tract and epithelial cells, rendering it a well-suited and widely utilized model for investigating TJs [49,50] and NDV pathogenesis [51,52]. Therefore, A549 cells were selected as the appropriate model for this study. We observed that following NDV infection, the TEER value decreases, whereas cell migration is enhanced. TEER serves as a key indicator for assessing the integrity and function of intercellular TJs; its decline directly signifies damage to these junctions [34]. Moreover, disruption of TJs can itself promote cell migration [53]. Based on these findings, we speculate that NDV likely disrupts the normal function of intercellular TJs. OCLN, ZO-1, and CLDN1 are classic TJ proteins crucial for maintaining TJs. Here, NDV significantly reduces the expression of all three proteins in A549 cells. We also tested NDV infection in other susceptible cells (HeLa, CTE, and HD11) and found it can generally lower most of TJ protein expression. Notably, we detected no expression of CLDN1 in HeLa or HD11 cells; while HeLa's lack of CLDN1 is known [35], this is a new finding for HD11. In contrast, OCLN and ZO-1 expression, though variable across cell types, generally shows a downward trend after infection. Epithelial cells are often the standard models for TJ studies, but some non-epithelial cells can also form TJ-like structures and express significant amounts of TJ proteins, making them potential research models as well [8,9]. This flexibility in model choice was particularly relevant for our work, as primary CTE cells presented practical challenges. Although HD11 cells are not epithelial, we observed that they possess TJ-like structures and exhibit stable expression levels of OCLN and ZO-1. After careful consideration, we selected HD11 cells, an NDV-susceptible macrophage cell line, for our validation model on avian derived cells.

Although the expression level of TJ proteins decreases, NDV infection does not result in their decreased mRNA levels. This observation suggests that the reduction in protein levels is unlikely due to diminished gene transcription, but rather reflects the action of protein degradation pathways. Currently, three major recognized protein degradation pathways are the caspase hydrolase pathway, the autophagy pathway, and the proteasome pathway [36]. Here, NDV induces TJ protein

degradation through multiple pathways, with the caspase hydrolase pathway being the most significant. As caspases are proteases that catalyze peptide bond hydrolysis and target key cellular proteins for degradation [54], this finding aligns with previous demonstrations that NDV triggers cell apoptosis [55]. This observation is also consistent with research on influenza infection. Gopal et al. have demonstrated that influenza virus triggers epithelial cell apoptosis, which then degrades TJ proteins [56]. NDV's infectivity and pathogenicity result from multiple viral proteins working together. Our study shows this complexity also applies to NDV's regulation of TJs. Specifically, the NP protein significantly reduces OCLN levels, while the F and HN proteins significantly lower ZO-1 levels. NP, F, and HN are the primary regulators of NDV infection and pathogenicity. The NP protein is abundant and conserved, essential for viral genome synthesis; F and HN are two membrane glycoproteins critical for infection, including attachment, entry, fusion, and release [22]. The regulation of TJs by NDV is complex, as their damage requires interactions among multiple viral proteins rather than the action of a single one. Importantly, we found that virulent NDV severely disrupts TJs, whereas avirulent NDV has little effect. This difference may primarily stem from distinct cellular responses: the virulent NDV replicates rapidly, exhibits high cytotoxicity, and efficiently induces apoptosis [57], with caspases degrading TJ proteins—a key destructive mechanism. In contrast, the avirulent NDV replicates more mildly and induces apoptosis poorly [58], insufficient to activate this pathway. Additionally, NDV-induced TJ disruption relies on the concerted action of multiple viral proteins (e.g., F and HN). Since virulent and avirulent NDVs differ in their viral proteins, particularly F and HN [59], these differences likely explain the distinct effects on TJs observed between the two strains. Moreover, only virulent NDV, not avirulent NDV, can downregulate TJ protein expression in the lungs of SPF chicken post-infection. The effect of NDV in disrupting TJ integrity has been validated in both *in vivo* and *in vitro* models, indicating a potential association with the pathogenic characteristics. Many other viruses, like influenza and porcine reproductive and respiratory syndrome virus (PRRSV), use this same strategy to boost their replication [37,38]. We confirmed that downregulating these proteins also enhances NDV replication. Interestingly, TJ proteins have less impact on NDV replication in the late stage compared to the early stage. We believe this is because, by the late stage, the virus has already peaked in replication; reduced cell viability and migration ability also lessen the difference between the knockdown group and the control group. Our previous study shows that NDV spreads rapidly within the host after infection, and this spread is crucial for its pathogenicity [26]. TJ proteins are known to inhibit viral spread. For instance, HPV and HAZV are limited by these proteins for diffusion [30,31]. We found that OCLN and ZO-1 can specifically restrict NDV spread. For NDV, the membrane fusion is an important pathway for its infection and pathogenicity, requiring the formation of a complex between the F and HN proteins. In this study, the disruption of TJ integrity by the F and HN proteins may be linked to their mediation of membrane fusion, a process that represents a key factor in promoting the further replication and spread of NDV. Future research is warranted to fully delineate the precise molecular mechanism. Therefore, NDV enhances both its replication and its ability to spread by downregulating TJ proteins.

TJ proteins maintain intercellular barrier integrity with the cytoskeleton, especially F-actin. ZO-1 acts as a scaffold, linking F-actin inside the cell to OCLN at the membrane, forming a strong TJ structure [60,61]. Our study demonstrates that NDV infection not only affects TJ protein expression but also disrupts TJ stability. *In vivo*, NDV causes loss of lung structure, disrupted alveolar junctions, and enhanced inflammatory levels. It is reported that severe inflammatory response can contribute to aggravating pulmonary histopathology by damaging intercellular TJs [62]. Therefore, TJ injury and inflammation act in synergy, mutually reinforcing each other. The elevated levels of inflammatory cytokines induced by the virulent NDV may, in turn, exacerbate the TJ injury. Concurrently, it induces the redistribution of both TJ structures and F-actin *in vitro*, while also triggering vimentin rearrangement. Intriguingly, during NDV infection, both vimentin and F-actin cytoskeletal components rearrange to form cage-like structures, which exhibit significant co-localization, suggesting a synergistic interaction. Similar cage-like structures have been observed in other RNA virus infections, such as those caused by SARS-CoV-2 [63], Zika [41], and PRRSV [42]. Notably, these structures often encapsulate viral components, facilitating their transport, assembly, and replication. Consistent with this, we found that the NDV NP protein and the TJ protein OCLN accumulate within these cage-like structures during infection. Inhibition of vimentin rearrangement not only

prevented cage-like structure formation but also significantly reduced NP protein levels and OCLN redistribution. Conversely, inducing vimentin rearrangement promotes these effects. Therefore, NDV-induced vimentin rearrangement is a key factor in TJ injury; its cage-like structure is closely associated with virus replication and TJ reorganization. However, the specific stage of NDV replication affected by the cage-like structure, and whether they influence NP-associated RNP complex formation, requires further investigation. This study also establishes for the first time a close link between the redistribution of vimentin/F-actin cage and TJ structures, warranting future mechanistic studies. The TJ integrity is crucial for viral spread [30,31]. Here, TJ proteins can negatively regulate NDV spread. Building on this, we further showed that TJ injury is necessary for effective NDV spread and even inflammatory response, a process directly regulated by vimentin rearrangement. Thus, NDV promotes its replication, spread, and inflammatory response by inducing vimentin

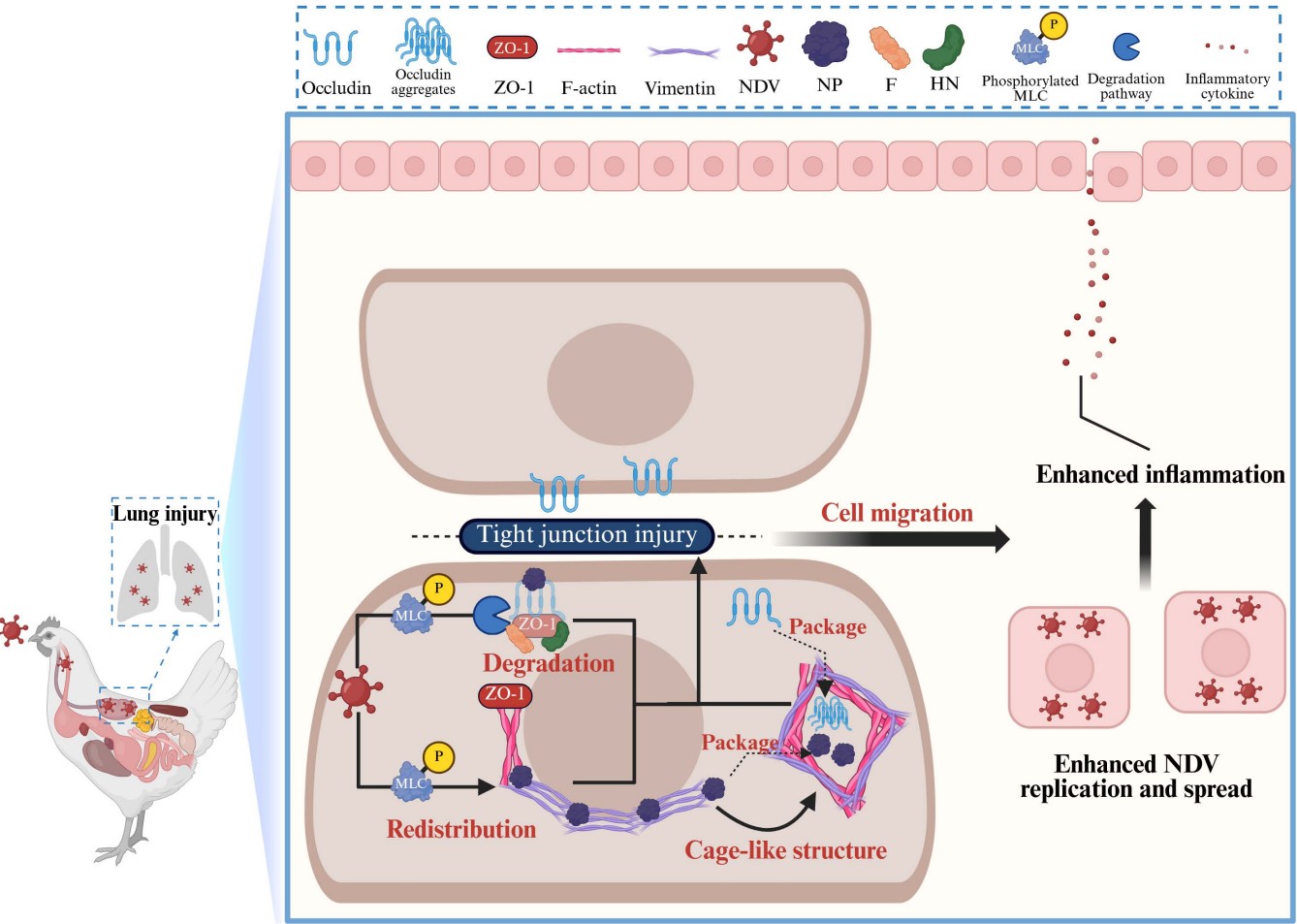

**Fig 7. Schematic diagram illustrating how NDV disrupts TJ injury by vimentin rearrangement, thereby promoting its spreading infection and inflammatory response.** TJs serve as a key barrier, crucial for maintaining epithelial homeostasis, resisting pathogens, and preventing viral dissemination. Following respiratory infection, NDV rapidly spreads within chickens, leading to acute lung injury. In cell biology, NDV infection disrupts TJ integrity and enhances infected cell migration, thereby promoting viral spread and infection. This effect is mediated by two primary mechanisms: first, downregulating TJ proteins (OCLN and ZO-1) through degradation pathways like caspase-mediated proteolysis and viral proteins (NP, F, HN), further disrupting TJs; second, inducing conformational changes in intermediate filaments (vimentin) and microfilaments (F-actin), leading to the formation of a cage-like structure that facilitates viral replication and TJ redistribution. Vimentin rearrangement plays a crucial role in NDV-induced TJ injury, promoting viral spread and inflammation, a process that is dependent on the activation of the MLC/p-MLC pathway. Created in BioRender. Lu, D. (2026) https://BioRender.com/o07c788.

rearrangement to disrupt TJs. Mechanistically, cytoskeletal structure adapts to external stress through activated signaling pathways. Among these, MLC phosphorylation (p-MLC) is a key mechanism boosting myosin contraction, which consequently disrupts TJ [5]. Consistent with this, our results show that NDV infection activates the MLC/p-MLC pathway, contributing to cytoskeleton rearrangement and ultimate TJ injury. Moreover, TJ injury has been demonstrated to promote cell migration during NDV infection, thereby accelerating virus replication and spread. It is reported that MLC/p-MLC not only influences cytoskeletal dynamics but its activation state is also sufficient to promote cell motility and migration [64]. Our study further confirms that MLC/p-MLC activation enhances the migration of infected cells, ultimately driving NDV replication and spread. Therefore, MLC/p-MLC appears to be an important target through which NDV utilizes vimentin-mediated TJ disruption to facilitate its own replication and spread.

In conclusion, this study demonstrates for the first time that NDV promotes viral replication and spread by activating MLC/p-MLC, and that vimentin-mediated TJ injury is a contributing factor (Fig 7). These findings exhibit broad-spectrum cellular characteristics, primarily observed in the virulent NDV, whereas the avirulent NDV fails to elicit a similar response. This investigation elucidates the molecular mechanism behind viral dissemination and acute lung injury induced by NDV, providing novel insights into the pathogenesis of viral pneumonias, including those caused by the novel coronavirus and influenza virus, and offering a foundation for the development of intervention strategies.

## Supporting information

**S1 Fig. Cell migration observed in the later stages after NDV infection.** A transwell assay was performed on NDV- and mock-infected A549 cells (MOI = 0.1) at 24 hpi. The assay involved seeding $4 \times 10^4$ cells in serum-free medium into the upper chamber, with the lower chamber supplemented with complete culture medium containing 20% fetal bovine serum to stimulate migration. Scale bar: 50 μm. *** P < 0.001; **** P < 0.0001.
(TIF)

**S2 Fig. Measurement of cell morphology and TJ protein levels post-NDV infection across diverse cell lines. (A–C)** The cellular morphology was monitored in NDV- and mock-infected HeLa, HD11, and CTE cells at an MOI of 1 from 6 to 24 hpi under light microscopy. Scale bar: 100 μm. **(D–E)** The expression levels of TJ-associated proteins OCLN and ZO-1 were assessed by western blotting at 24 hpi with NDV or mock (MOI = 1). The gray value of each protein was quantified by Image J and normalized to ACTB. The gray value of mock-infected group was considered as "1", respectively.
(TIF)

**S3 Fig. Measurement of CLDN1 expression levels post-NDV infection across diverse cell lines.** The expression levels of CLDN1 protein were assessed by western blotting at 24 hpi with NDV or mock (MOI = 1) in **(A)** A549, **(B)** CTE, **(C)** HeLa, and **(D)** HD11 cells. The gray value of each protein was quantified by Image J and normalized to ACTB. The gray value of mock-infected group was considered as "1", respectively.
(TIF)

**S4 Fig. The mRNA levels of TJ genes in NDV-infected A549 cells.** The mRNA levels of OCLN and ZO-1 genes were evaluated by qPCR following infection with NDV at 1 MOI for 24 h. NDV-infected groups were compared to the mock-infected group, and statistical analysis was carried out. ** $P < 0.01$; *** $P < 0.001$; **** $P < 0.0001$.
(TIF)

**S5 Fig. Downregulation of TJ proteins enhance viral replication and spread in HD11 cells.** siRNA-OCLN and siRNA-ZO-1 (60 pmol) were employed to generate HD11/OCLN-KD and HD11/ZO-1-KD cells, respectively. siRNA-NC transfected cells were recognized as the control HD11/ctrl. **(A)** OCLN protein levels and cell viability were determined after siRNA transfection. The transfected groups were compared to the control group, and statistical analysis was performed. **(B)** ZO-1 protein levels and cell viability were determined after siRNA transfection. The transfected groups were compared to the

control group, and statistical analysis was performed. **(C)** HD11/ctrl and HD11/OCLN-KD cells were infected with NDV at 1 MOI for 24 h. The viral replication in cells was evaluated by western blotting. The gray value of each protein was quantified by Image J and normalized to ACTB. The gray value of NDV-infected HD11/ctrl group was considered as "1". Subsequently, these cells were infected with NDV at 0.1 MOI for 12, 24, and 36 h. The viral load in supernatants was detected by $TCID_{50}$. **(D)** HD11/ctrl and HD11/ZO-1-KD cells were infected with NDV at 1 MOI for 24 h. The viral replication in cells was evaluated by western blotting. The gray value of each protein was quantified by Image J and normalized to ACTB. The gray value of NDV-infected HD11/ctrl group was considered as "1". Subsequently, these cells were infected with NDV at 0.1 MOI for 12, 24, and 36 h. The viral load in supernatants was detected by $TCID_{50}$. **(E)** HD11/ctrl, HD11/OCLN-KD and HD11/ZO-1-KD cells were infected with NDV at 0.00001 MOI for 24 h, and the foci in cells were evaluated by IFA. **(F)** Fifteen random lesions per treatment group were chosen, and the extent of viral dissemination was determined by counting the number of cells within each lesion. Scale bar: 50 μm. * $P < 0.05$; ** $P < 0.01$; *** $P < 0.001$; **** $P < 0.0001$; ns, no significant difference.
(TIF)

**S6 Fig. The mRNA levels of inflammatory genes in lung tissue.** 4-week-old SPF chickens were infected with $10^5$ $EID_{50}$ NDV via the intranasal and intraocular route for 4 days. PBS group was considered as the control group. Lung tissue was then collected to measure the mRNA levels of inflammatory genes using qPCR. *** $P < 0.001$; **** $P < 0.0001$; ns, no significant difference.
(TIF)

**S7 Fig. La Sota infection-induced alterations in TJ proteins and pathological changes in chicken lung tissues.** 4-week-old SPF chickens were infected with $10^5$ $EID_{50}$ La Sota via the intranasal and intraocular route for 4 days. PBS group was considered as the control group. **(A)** The lung tissue was harvested to detect OCLN and ZO-1 expression levels by western blotting. The gray value of each protein was quantified by Image J and normalized to ACTB. The gray value of mock-infected group was considered as "1". **(B)** The histopathological alterations in chicken lung tissue post-La Sota infection were meticulously examined under 40× and 100× magnification using H&E staining techniques.
(TIF)

**S8 Fig. Structural examination of TJ complexes and vimentin networks following La Sota infection.** A549 cells were infected with La Sota at an MOI of 0.1 for 18 h. Mock group as a control for comparison. The localization of OCLN, ZO-1, F-actin **(A)**, and vimentin **(B)** was observed by confocal microscopy. Scale bar: 10 μm.
(TIF)

**S9 Fig. NDV replication is required for the disruption of TJs. (A)** OCLN and ZO-1 protein levels were assessed in A549 cells infected with either live NDV or UV-inactivated NDV (MOI = 1) by western blotting. The gray value of each protein was quantified by Image J and normalized to ACTB. The gray value of mock group was considered as "1". **(B)** The localization of OCLN, ZO-1, vimentin, and F-actin was observed in A549 cells infected with either live NDV or UV-inactivated NDV at 0.1 MOI by confocal microscopy. Scale bar: 10 μm.
(TIF)

**S10 Fig. Assessment of inflammatory cytokine mRNA levels in NDV-infected A549 cells post-IDPN or ajoene treatment.** The mRNA levels of inflammatory genes (IL-1β, IL-6, and TNF-α) in NDV-infected A549 cells (MOI = 0.01) were detected by qPCR, comparing levels induced by IDPN or ajoene treatment. ** $P < 0.01$; *** $P < 0.001$; **** $P < 0.0001$.
(TIF)

**S11 Fig. Structural observation of TJ and cytoskeleton structures in HD11 cells.** The structure of OCLN, ZO-1, F-actin, and vimentin was observed in normal HD11 cells.
(TIF)

**S12 Fig. Toxicity determination of IDPN in HD11 cells. (A)** Cytotoxicity of IDPN was assessed using the CCK8 assay. **(B)** Structural observation of vimentin in HD11 cells following IDPN treatment for 24 h. Scale bar: 10 µm. *** $P < 0.001$; **** $P < 0.0001$.
(TIF)

**S13 Fig. Inhibition of vimentin rearrangement attenuates TJ injury, thereby inhibiting viral replication and spread.**
**(A)** The structure of OCLN, ZO-1, F-actin, and vimentin was observed in NDV-infected HD11 cells (MOI = 0.1) at 18 hpi following IDPN treatment (1.5%) by confocal microscopy. The white arrows represent the vimentin and F-actin cages, whereas the dashed box highlights the OCLN aggregates in vimentin cages. Scale bar: 10 µm. **(B)** The replication levels of NDV (MOI = 0.1) were quantified by western blotting at 24 hpi, comparing NP levels in the presence and absence of IDPN (1.5%). The gray value of each protein was quantified by Image J and normalized to ACTB. The gray value of NDV-infected group was considered as "1". **(C)** The replication levels of NDV (MOI = 0.1) were quantified using the $TCID_{50}$ assay at 12 and 24 hpi, comparing the viral titers in the presence and absence of IDPN (1.5%). **(D)** The cell migration by NDV (MOI = 0.1) were assessed at 6 hpi using the transwell assay, comparing the amount of migrated HD11 cells in the presence and absence of IDPN (1.5%). Scale bar: 50 µm. **(E)** The cell foci were measured in NDV-infected (MOI = 0.00001) HD11 cells at 36 hpi by IFA, comparing the number of cells within foci in the presence and absence of IDPN (1.5%). Scale bar: 50 µm. *** $P < 0.001$; **** $P < 0.0001$.
(TIF)

**S14 Fig. Activation of vimentin rearrangement enhances TJ injury, thereby acilitating viral replication and spread. (A)** Cytotoxicity of ajoene was assessed using the CCK8 assay. Structural examination of vimentin, F-actin, OCLN, and ZO-1 in HD11 cells treated with ajoene (10 µM) was conducted. The white arrow indicates the rearranged structure. Scale bar: 10 µm. **(B)** The NP protein levels were examined in NDV-infected (MOI = 1) HD11 cells treated with or without ajoene (10 µM) by western blotting. The gray value of each protein was quantified by Image J and normalized to ACTB. The gray value of NDV-infected group was considered as "1". **(C)** The replication levels of NDV (MOI = 0.1) were quantified using the $TCID_{50}$ assay in HD11 cells, comparing the viral titers in the presence and absence of ajoene (10 µM). **(D)** The cell migration by NDV (MOI = 0.1) were assessed at 6 hpi using the transwell assay in HD11 cells, comparing the amounts of migrated cells in the presence and absence of ajoene (10 µM). Scale bar: 50 µm. **(E)** The cell foci were measured in NDV-infected HD11 cells (MOI = 0.00001) by IFA, comparing the number of cells within foci in the presence and absence of ajoene (10 µM). Scale bar: 50 µm. * $P < 0.05$; ** $P < 0.01$; *** $P < 0.001$; **** $P < 0.0001$.
(TIF)

**S15 Fig. Assessment of inflammatory cytokine mRNA levels in NDV-infected HD11 cells post-IDPN or ajoene treatment.** The mRNA levels of inflammatory genes (IL-1β, IL-6, and TNF-α) in NDV-infected HD11 cells (MOI = 0.01) were detected by qPCR, comparing levels induced by IDPN (1.5%) or ajoene (10 µM) treatment. * $P < 0.05$; *** $P < 0.001$; **** $P < 0.0001$.
(TIF)

**S1 Table. Characteristics of NDV strains used in this study.**
(DOCX)

**S2 Table. Primer sequences designed for constructing eukaryotic expression plasmids of viral proteins.**
(DOCX)

**S3 Table. siRNA sequences designed for the specific knockdown of OCLN and ZO-1 gene.**
(DOCX)

**S4 Table. Primers sequences used in qPCR assay.**
(DOCX)

**S5 Table. Virus shedding in oropharyngeal and cloacal swabs of chickens, determined by inoculating SPF chicken embryos.**
(DOCX)

## Acknowledgments

We are especially grateful to Prof. Chan Ding (Shanghai Jiaotong University) for his guidance in the study design. We are also grateful to all colleagues for their insightful input and numerous valuable suggestions.

## Author contributions

**Conceptualization:** Min Gu.

**Data curation:** Yu Chen.

**Formal analysis:** Ruyi Gao.

**Funding acquisition:** Xiaolong Lu, Kaituo Liu, Shunlin Hu, Xiufan Liu, Xiaowen Liu.

**Investigation:** Xiaolong Lu, Qiwen Zhou, Meiqi Li.

**Methodology:** Jiao Hu.

**Project administration:** Xinan Jiao, Xiaoquan Wang, Xiufan Liu, Chan Ding, Xiaowen Liu.

**Software:** Wenhao Yang.

**Supervision:** Shunlin Hu.

**Validation:** Mingzhu Wang.

**Visualization:** Tianxing Liao.

**Writing – original draft:** Xiaolong Lu.

**Writing – review & editing:** Xiaoquan Wang, Chan Ding.

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
