## [Decision Letter · Decision Letter 0]

10 Apr 2025

Newcastle disease virus facilitates viral replication and spread through vimentin-dependent tight junction injury

PLOS Pathogens

Dear Dr. Liu,

Thank you for submitting your manuscript to PLOS Pathogens. After careful consideration, we feel that it has merit but does not fully meet PLOS Pathogens's publication criteria as it currently stands. Therefore, we invite you to submit a revised version of the manuscript that addresses the points raised during the review process.

Please submit your revised manuscript within 60 days Jun 09 2025 11:59PM. If you will need more time than this to complete your revisions, please reply to this message or contact the journal office at plospathogens@plos.org. Please include the following items when submitting your revised manuscript:

We look forward to receiving your revised manuscript.

Kind regards,

Maria João Amorim, Ph.D

Academic Editor

PLOS Pathogens

Michael Letko

Section Editor

PLOS Pathogens

Editor-in-Chief

PLOS Pathogens

orcid.org/0000-0003-2946-9497

Editor-in-Chief

PLOS Pathogens

orcid.org/0000-0002-7699-2064

**Additional Editor Comments:**

Dear Dr. Xiaowen Liu,

Thank you for your patience during the peer review process for your manuscript, "Newcastle disease virus facilitates viral replication and spread through vimentin-dependent tight junction injury." The manuscript has now been evaluated by reviewers with relevant expertise.

As detailed in their comments below, the reviewers raises significant concerns regarding the conclusiveness of the study. In light of these evaluations, we regret to inform you that we can only consider a revised version of your manuscript, provided that all reviewers’ concerns are thoroughly addressed.

We acknowledge that addressing these points will require substantial additional work, and we understand that this may lead to modifications of your current conclusions. As such, we cannot guarantee acceptance of the revised manuscript.

The reviewers highlighted several major issues that must be addressed for further consideration:

Model choice: There are concerns about the use of A549 epithelial cells and HD11 chicken macrophages as infection models. Please either justify the use of these models or consider repeating key experiments using more physiologically relevant systems, as NDV does not infect humans and macrophages' TJ are limited.

Discussion section: The current discussion largely reiterates the results and does not adequately explore the mechanisms behind vimentin-mediated tight junction damage. It would benefit from a substantial rewrite to focus on interpretation, underlying mechanisms, and broader implications.

Mechanistic insight: The manuscript does not identify which viral proteins or infection processes are responsible for the observed vimentin-mediated TJ injury. Clarifying this point would significantly strengthen your findings.

Additional TJ markers: Inclusion of data on other tight junction proteins would enhance the robustness of your conclusions.

Central claim: Most critically, the manuscript does not convincingly establish a causal link between vimentin-mediated TJ injury and enhanced NDV replication and transmission. This point must be addressed with additional evidence or revised interpretations.

We hope these comments help guide your decision about how to proceed. Should you choose to undertake the necessary revisions, please add the reference PPATHOGENS-D-24-02713 to facilitate the choice of academic editor and the review process.

**Journal Requirements:**

1) Please provide an Author Summary. This should appear in your manuscript between the Abstract (if applicable) and the Introduction, and should be 150-200 words long. The aim should be to make your findings accessible to a wide audience that includes both scientists and non-scientists. Sample summaries can be found on our website under Submission Guidelines:

https://journals.plos.org/plospathogens/s/submission-guidelines#loc-parts-of-a-submission

3) Some material included in your submission may be copyrighted. According to PLOSu2019s copyright policy, authors who use figures or other material (e.g., graphics, clipart, maps) from another author or copyright holder must demonstrate or obtain permission to publish this material under the Creative Commons Attribution 4.0 International (CC BY 4.0) License used by PLOS journals. Please closely review the details of PLOSu2019s copyright requirements here: PLOS Licenses and Copyright. If you need to request permissions from a copyright holder, you may use PLOS's Copyright Content Permission form.

Potential Copyright Issues:

- Figures 1, 3, and 9. Please confirm whether you drew the images / clip-art within the figure panels by hand. If you did not draw the images, please provide (a) a link to the source of the images or icons and their license / terms of use; or (b) written permission from the copyright holder to publish the images or icons under our CC BY 4.0 license. Alternatively, you may replace the images with open source alternatives. See these open source resources you may use to replace images / clip-art:

4) Please ensure that the funders and grant numbers match between the Financial Disclosure field and the Funding Information tab in your submission form. Note that the funders must be provided in the same order in both places as well.

**Reviewers' Comments:**

Reviewer's Responses to Questions

**Part I - Summary**

Reviewer #1: In “Newcastle disease virus facilitates viral replication and spread through vimentin dependent tight junction injury,” Lu et al. provide experimental results showing that NDV infection disrupts TJ protein expression of OCLN and ZO-1, which have previously been implicated in intercellular contact. The authors have performed many experiments to show not only that these genes are associated with NDV infection, but that they are important players to increase the pathogen’s virulence within hosts. The experiments include: (1) a transwell assay showing the difference in cell migration between knockdown cells and control, (2) a few experiments (both in-vitro and in-vivo) to implicate OCLN and ZO-1 in cell migration, showing that downregulation decreases cell migration, and viral shedding (3) several experiments, including a microscopy study, to observe TJ structural differences of the vimentin and F-actin fiber between knockdown cells and control. They also provide a lengthy discussion on the pivotal role of vimentin on viral replication. Overall, an impressive amount of interesting work has been done to elucidate the exact mechanisms of NDV infection and virulence within poultry. My recommendation is for a minor revision to further clarify several areas where the text was slightly unclear.

Reviewer #2: The authors have studied tight junction injury during NDV infection. They have used lentogenic and velogenic NDV strains in infection models in A549 cells (human epithelial cell line), HD11 cells (chicken macrophage cell line) and chickens. They conclude that NDV replication and spread is facilitated by vimentin-mediated TJ injury.

Virulence of NDV is predominantly determined by the cleavage site of the fusion protein: the F protein of velogenic contains a polybasic cleavage, and can be cleaved by proteases that are present in different tissues. However, other factors will likely contribute to NDV virulence. Surprisingly, the authors do not summarize current knowledge on NDV virulence in either introduction or discusison.

The manuscript contains a large dataset, with a strong focus on analysis of processes that are observed in cells infected with velogenic NDV. However, in my opinion the data do not support a causal relationship between vimentin-mediated TJ injury and NDV replication and spread. Most importantly, the authors do not provide a biologically plausible mechanism by which NDV causes vimentin-mediated TJ injury. Therefore, it is unclear if the observed TJ injury is either cause or effect of velogenic NDV infection. In my opinion, a “precise mechanism governing viral dissemination” (line 26) is not provided. In addition, it is unclear to what extent the observations are specific for NDV.

Reviewer #3: This manuscript thorough described the mechanism by which virulent NDV infection induce vimentin-dependent tight junction injury of rearranging vimentin and F-action, which resulted in the degradation of the TJ proteins OCLN and ZO-1. The author conducted in vitro experiments in both human lung epithelial cells and chicken macrophages. In vivo infection of chickens with NDV also showed disruption of TJ of lung epithelial cells. The studies are very well conducted and manuscript is well-written. There are only a few minor comments need to be addressed.

**Part II – Major Issues: Key Experiments Required for Acceptance**

Reviewer #1: Major Comments:

L354: What was the rationale for choosing OCLN and ZO-1 protein levels as opposed to other TJ associated proteins?

L372 and L523: One thing that slightly confused me here and in L523 was how exactly the apoptotic pathway is being investigated, when it appears to be an intracellular pathway rather than an intercellular pathway, which was the focus of many of the experiments. It would also be good to identify what these “diverse” pathways may be that the authors write. Furthermore, is there a link between the apoptotic pathway and the cage formation beginning in L447? Clearer links between these results would benefit the clarity of the text.

L399: Could you describe in more detail the implications of the time-dependence of the difference in infection? As the viral titer is a cumulative sum, does this mean that there is no discernible difference in the overall concentration of virus proteins (such as NP) between the knockdown cells and control cells eventually? Does this mean that the knockdown cells migrate to more tissues early in the infection, and this is what makes a difference in overall pathogenicity and virulence of infection, even though the overall concentration of virus proteins will eventually be the same?

L512: I am still confused about the significance of observation and F-actin formed a cage around the NP and OCLN aggregates. What makes them “factories”? Is there some sort of input and output of these aggregates?

L829: Have other studies been done on other TJ proteins, or other intercellular or intracellular proteins? It would be useful to contextualize these results with other research to understand if they are the sole drivers of NDV infection and pathogenicity, or if there are others.

Reviewer #2: in my opinion this paper does not elucidate a molecular mechanism (as claimed at the end of the discussion), but describes in detail the cytopathic changes in cells that are induced by velogenic NDV infection. There is no evidence for a specific NDV interaction with vimentin or with TJs.

Reviewer #3: none

**Part III – Minor Issues: Editorial and Data Presentation Modifications**

Reviewer #1: Minor Comments:

L71: Should be “However, the infected cells can also be…”?

L380: Should be “disrupting” instead of “distrusting”?

L456: Not sure why the sentence starts with “Meanwhile,” – maybe “In addition,” would work better? Same with L523

L789: Eliminate “preliminarily”

Reviewer #2: 1. Lines 49-51: the authors suggest that all cells contain TJs. TJs play a crucial role in epithelial barriers (as mentioned in line 51). However, the authors do not discuss the presence, absence or importance of TJs in non-epithelial cells.

2. It has been described that TJ formation in A549 cells is inconsistent (Radiom et al., PMID 33173147). Therefore, it is surprising that the authors have selected this human cell line as their most important model. Velogenic NDV strains do not cause disease in (immunocompetent) humans. Moreover, the second (avian) cell line is a macrophage cell line, a cell type in which TJs play a limited role. The infection models used by the authors should be better explained and justified.

3. Line 98: reference 27 does not describe virus infections, while reference 28 describes TJ damage in the context of breaching the blood-brain barrier. The references do not support the statement in the sentence.

4. Lines 159-169: the authors use a transwell system, but do not specify which transwells were used (what material?) and do not specify the pore size. The principle of the assay migration assay is not clear to me. If this is an assay that has been used before, the authors should provide a reference. With the limited description in this paragraph the assay cannot be reproduced, and the results cannot be interpreted.

5. Lines 382-383: the data do not support a causal relationship as suggested by the paragraph title. In addition, it is unclear if the observations are specific for NDV.

6. Is the cage formation described by the authors related to inclusion bodies as described by others? As far as I can see the authors do not link this observation to similar observations in the literature.

7. Lines 584-585: it is impossible to study inflammation in an in vitro model that lacks immune cells.

8. Line 617: the background and mechanism of ajoene as vimentin rearrangememnt activator should be explained and cited in the manuscript.

9. Lines 660-664: why do the authors use HD11 cells (a chicken macrophage line) and not an avian epithelial cell line? Macrophages do not normally form tight junctions.

10. The discussion is long, but is mostly a reiteration of the results of the study. Most importantly, the authors do not discuss what viral proteins, viral factors infection processes cause the vimentin-mediated TJ damage (see eg lines 838-839).

11. Line 863: NDV virons are formed by budding, so it is unlikely that thes accumulate within cytoskeletal cages. Do the authors mean RNPs?

12. Lines 888-889: the model presented in figure 9 is not convincing. The term “through” in line 889 is not supported by the data.

Reviewer #3: -Line 71, “the infected cells can be also susceptible to entry into the bloodstream following shedding” this sentence is not clear.

-Statement in line 395 related to Figure 2F, the increase of NDV protein relative to the ACTB is not prominent as there is also increase in ACTB level.

-line 456, re-phrase significantly as no statistical significance analysis was performed.

-Treatment of mock infected cells with IDPN should be included

-Line 602, the observation of “cell number within the viral foci in the normal infection group” should be defined as syncytia formation, as cells fused together via the expression of viral protein to form a multinuclear cell. Therefore it’s not so accurate to describe each nucleus as a single cell. Same with line 630 figure 6G, 7J and 8D and 8I

-Figure 6B, it doesn’t seems to change the distribution of F-actin.

-Please discuss more of the mechanism of rearrangement of vimentin by viral infection. Such as if/which viral protein, not limited to NDV, interact with vimentin and why action cage can enhance viral replication.

-Please discuss why avirulent NDV doesn’t cause vimentin-dependent TJ injury as much as the virulent NDV does.

PLOS authors have the option to publish the peer review history of their article (what does this mean? ). If published, this will include your full peer review and any attached files.

**Do you want your identity to be public for this peer review?** For information about this choice, including consent withdrawal, please see our Privacy Policy .

Reviewer #1: No

Reviewer #2: No

Reviewer #3: No

**Figure resubmission:**

**Reproducibility:**



---

## [Decision Letter · Decision Letter 1]

15 Aug 2025

Dear Prof. Liu,

We are pleased to inform you that your manuscript 'Newcastle disease virus promotes spreading infection through vimentin-dependent tight junction injury mediated by MLC/p-MLC activation' has been provisionally accepted for publication in PLOS Pathogens.

Before your manuscript can be formally accepted please consider the last comments of the reviewers and you will need to complete some formatting changes, which you will receive in a follow up email. A member of our team will be in touch with a set of requests.

Best regards,

Maria João Amorim, Ph.D

Academic Editor

PLOS Pathogens

Michael Letko

Section Editor

PLOS Pathogens

Sumita Bhaduri-McIntosh

Editor-in-Chief

PLOS Pathogens

orcid.org/0000-0003-2946-9497

Michael Malim

Editor-in-Chief

PLOS Pathogens

orcid.org/0000-0002-7699-2064

Reviewer Comments (if any, and for reference):

Reviewer's Responses to Questions

**Part I - Summary**

Reviewer #1: The authors have addressed all major and minor comments I suggested. Below is a review of changes that have been made in response to my major comments, as well as two minor typos that should be corrected.

Review:

The authors have added a clarification to highlight that differences between knockdown and control cells are less pronounced in later stages of infection (L925-929) as well as reasons for why these differences may be less pronounced later on.

The authors add a more-detailed description of the F-actin cage formation of vimentin rearrangement after NDV infection (L), especially referencing that the “cage-like structure is closely *associated* with virus replication and TJ reorganization” and highlight the important need for testing for the exact causal role of the cages on viral replication (L944-961).

The authors add references and further description to describe the apoptotic pathway used for cellular degradation by NDV infection (L893-901).

The authors provide some context and supplementary figures on another TJ protein (CLDN1) that was not used for further analysis in this study, as well as justification for why it was not used, and report some findings of their preliminary analyses of CLDN1 (L875-887).

Reviewer #2: The authors have prepared a detailed rebuttal, in which most of the comments of the reviewers are adequately addressed. In addition, they have performed new experiments and added relevant references in the revised manuscript. In my opinion this has substantially improved the quality of the manuscript. Most importantly, some of the conclusions have been modified or rephrased.

Reviewer #3: The authors addressed all my minor comments

**Part II – Major Issues: Key Experiments Required for Acceptance**

Reviewer #1: N/A

Reviewer #2: I have one remaining but crucial criticism to the study. Although the authors now acknowledge that the cleavage site of F plays an important role in NDV virulence (line 98), this important biological property is not discussed as a major contributor to NDV spread both within and between hosts. Several studies have shown that modification of the polybasic cleavage site of velogenic NDV strains to a monobasic cleavage site results in a transition to a mesogenic or lentogenic phenotype, which will no doubt be associated with reduced TJ injury (both in vitro and in vivo). From that perspective, it could be concluded that biological activity of the NDV F protein (i.e. cleaved or non-cleaved, in presence of a functional HN protein) is likely is the most important determinant of the level of TJ injury. Nevertheless, the authors do not discuss the role of membrane fusion in the impact of NDV on TJ integrity. The role of NDV F in the virulence, TJ injury and spread should be addressed more explicitly in the discussion.

Reviewer #3: (No Response)

**Part III – Minor Issues: Editorial and Data Presentation Modifications**

Reviewer #1: Minor edits:

L606: “This finding suggests *their* potential roles as hubs for viral replication and TJ redistribution”

L957-958: “However, the specific stage of NDV replication *affected* by the cage-like structure.”

Reviewer #2: N/A

Reviewer #3: (No Response)

PLOS authors have the option to publish the peer review history of their article (what does this mean? ). If published, this will include your full peer review and any attached files.

**Do you want your identity to be public for this peer review?** For information about this choice, including consent withdrawal, please see our Privacy Policy .

Reviewer #1: No

Reviewer #2: No

Reviewer #3: No

---

## [Editor Report · Acceptance letter]

Dear Prof. Liu,

We are delighted to inform you that your manuscript, " 

Newcastle disease virus promotes spreading infection through vimentin-dependent tight junction injury mediated by MLC/p-MLC activation," has been formally accepted for publication in PLOS Pathogens.

Best regards,

Sumita Bhaduri-McIntosh

Editor-in-Chief

PLOS Pathogens

orcid.org/0000-0003-2946-9497

Michael Malim

Editor-in-Chief

PLOS Pathogens

orcid.org/0000-0002-7699-2064